# Local Covariate Selection for Average Causal Effect Estimation without Pretreatment and Causal Sufficiency Assumptions

Zeyu Liu[1]  Zheng Li[2 3]  Feng Xie[1]  Yan Zeng[1]  Hao Zhang[2]  Kun Zhang[4 5]

## Abstract

We study the problem of selecting covariates for unbiased estimation of the total causal effect. Existing approaches typically rely on global causal structure learning over all variables, or on strong assumptions such as causal sufficiency—where observed variables share no latent confounders—or the pretreatment assumption, which limits covariates to those unaffected by the treatment or outcome. These requirements are often unrealistic in practice, and global learning becomes computationally prohibitive in high-dimensional settings. To address these challenges, we propose a novel local learning method for covariate selection in nonparametric causal effect estimation that avoids both the pretreatment and causal sufficiency assumptions. We first characterize a local boundary that contains at least one valid adjustment set whenever one exists for identifying the causal effect, and then develop local identification procedures to efficiently search within this boundary. We prove that the proposed method is sound and complete. Experiments on multiple synthetic datasets and two real-world datasets show that our approach achieves accurate causal effect estimation while substantially improving computational efficiency.

## 1. Introduction

Causal effect estimation is central to understanding the impact of interventions across domains such as medicine (Ru-

[1]Department of Applied Statistics, Beijing Technology and Business University, Beijing, China [2]Shenzhen Institutes of Advanced Technology, Chinese Academy of Sciences, Shenzhen, China [3]College of Computer Science and Artificial Intelligence, Fudan University, Shanghai, China [4] Carnegie Mellon University, Pittsburgh, PA, USA  [5]Department of Machine Learning, Mohamed bin Zayed University of Artificial Intelligence, Abu Dhabi, UAE. Correspondence to: Feng Xie <fengxie@btbu.edu.cn>, Yan Zeng <yanazeng013@btbu.edu.cn>.

*Proceedings of the 43rd International Conference on Machine Learning*, Seoul, South Korea. PMLR 306, 2026. Copyright 2026 by the author(s).

bin, 1974), economics (Angrist et al., 1996), social sciences (Spirtes et al., 2000; Pearl, 2009), and public policy (Imbens & Rubin, 2015; Hernán & Robins, 2020). It addresses questions of the form: What would happen to the outcome $Y$ if we intervened on the treatment $X$? A widely used strategy for causal effect estimation is covariate adjustment, which seeks a subset of observed variables—known as an adjustment set—to remove confounding bias (Pearl, 1993; Maathuis & Colombo, 2015; Perković et al., 2018). When the full causal structure is known, valid adjustment sets can be identified using graphical criteria such as the back-door criterion (Pearl, 1993; 2009) and its generalizations (Maathuis & Colombo, 2015; Perković et al., 2018). However, in practice, the underlying causal structure is rarely known in advance.

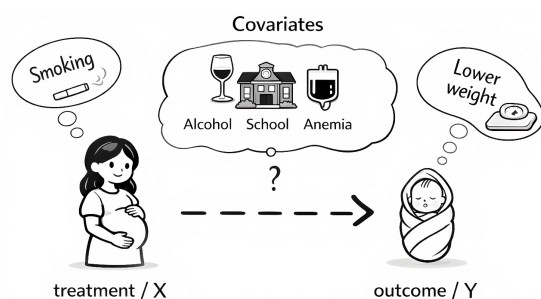

*Figure 1.* Illustration of the causal effect of maternal smoking (treatment $X$) on birth weight (outcome $Y$), with the covariates (e.g., alcohol use, education, and anemia) potentially affecting the $X \rightarrow Y$ relationship (Almond et al., 2005).

Consider, for example, the effect of maternal smoking during pregnancy on infant birth weight, illustrated in Figure 1. Besides the treatment and outcome, many additional variables—such as alcohol use, maternal education, and health conditions like anemia—may influence both smoking behavior and birth outcomes. To obtain an unbiased estimate of the causal effect, one must carefully select a set of covariates that blocks spurious back-door paths without introducing bias through inappropriate adjustment. In realistic settings, however, it is often unclear which variables are safe to adjust for, especially when some covariates may be affected by the treatment or share unobserved confounders with other variables.

*Figure 2.* Example mixed ancestral graph (MAG) illustrating the processing of $X$ and the resulting outcome $Y$. Red-shaded nodes denote valid adjustment sets. (a) Adapted from Cheng et al. (2022): CEELS fails to select any adjustment set, despite the presence of the COSO variable $V_1$. (b) The LSAS method, when the pretreatment assumption is not assumed, fails to identify a valid adjustment set within $MB(Y)$. (c) The Local Partition Discovery (LDP) method (Maasch et al., 2024) may fail to identify valid adjustment sets in certain cases, indicating a lack of completeness. (d) The LDP method fails to identify an adjustment set because its sufficient condition is violated, even though the pretreatment assumption holds. Moreover, this example contains no COSO variable, yet a valid adjustment set can still be found locally; in contrast, CEELS fails to select any adjustment set.

Several data-driven approaches have been proposed to estimate causal effects when the underlying causal graph is unknown. A representative method is IDA (Maathuis et al., 2009), which first learns a CPDAG using the PC algorithm (Spirtes & Glymour, 1991), enumerates DAGs in the corresponding Markov equivalence class, and estimates a multiset of possible causal effects. Subsequent extensions (Perkovic et al., 2017; Fang & He, 2020; Fang et al., 2025) incorporate background knowledge, such as partial edge directions or ancestral relations, to improve estimation precision. Related approaches, such as CovSel (Häggström et al., 2015), perform covariate selection based on graphical criteria (De Luna et al., 2011). However, the validity of these methods typically relies on the assumption of causal sufficiency, which requires that all common causes of observed variables are measured. This assumption is often violated in practice, where latent confounders are prevalent.

To relax causal sufficiency, LV-IDA (Malinsky & Spirtes, 2017) extends IDA to settings with latent variables using the generalized back-door criterion (Maathuis & Colombo, 2015). Subsequent work has developed more efficient procedures for handling latent confounding (Hyttinen et al., 2015; Wang et al., 2023a; Cheng et al., 2023). Nevertheless, these approaches still rely on learning a global causal graph. Such global learning can be computationally expensive and unnecessary when the goal is to estimate the causal effect between a specific treatment–outcome pair.

Entner et al. (2013) proposed EHS for covariate selection without learning the full causal structure. However, EHS performs an exhaustive search over all variables, which becomes computationally infeasible in high-dimensional settings. Subsequent work, such as CEELS (Cheng et al., 2022), improves efficiency through local search, but may miss valid adjustment sets that would be identified by a global search, as illustrated in Figure 2(a). LSAS (Li et al., 2025b) was later introduced as a complete method under this framework. Despite these advances, existing local covariate selection methods rely on the pretreatment assumption—that neither the treatment nor the outcome causally

influences the covariates. This assumption is often unrealistic and can lead to incorrect conclusions; for example, LSAS may fail to identify valid adjustment sets when the pretreatment condition is violated, as shown in Figure 2(b).

More recently, Maasch et al. (2024; 2025) proposed the Local Discovery by Partitioning (LDP) method, which relaxes the pretreatment assumption and identifies valid adjustment sets under certain sufficient conditions. However, LDP is not complete: as shown in Figure 2(d), it may fail to return a solution when these conditions are violated, even in cases where the pretreatment assumption actually holds.

*Table 1.* Comparison of local covariate selection methods. Here, *soundness* means that every selected adjustment set is valid for identifying the target causal effect (no false inclusions); and *completeness* means that the method returns a valid adjustment set whenever one is identifiable from data (no essential covariates missing).

| Method (local) | Soundness | Completeness | Without Pretreatment |
|---|---|---|---|
| CEELS | ✓ | ✗ | ✗ |
| LSAS | ✓ | ✓ | ✗ |
| LDP | ✓ | ✗ | ✓ |
| **LCS (Ours)** | ✓ | ✓ | ✓ |

These limitations highlight the need for a sound and complete local method for adjustment set selection under standard causal assumptions. As summarized in Table 1, existing local methods either rely on the pretreatment and causal sufficiency assumptions or are incomplete. Our method fills this gap, and we make the following contributions:

- We establish a theoretical characterization of a local boundary that contains all valid adjustment sets without relying on the pretreatment or causal sufficiency assumptions, and develop local identification rules to search within this boundary.
- Building on these local identification rules and local structure learning techniques, we propose a data-driven algorithm, **LCS**, for identifying adjustment sets for the

target causal effect, and prove that it is sound and complete.

- We demonstrate the effectiveness and computational efficiency of LCS on multiple synthetic datasets and two real-world datasets.

## 2. Preliminaries

### 2.1. Graph Terminology and Notations

Throughout this paper, we denote sets in bold uppercase letters (e.g., $\mathbf{V}$), graphs in calligraphic font (e.g., $\mathcal{G}$), and nodes or variables in uppercase letters (e.g., $V$).

**Directed Graph.** A graph $\mathcal{G} = (\mathbf{V}, \mathbf{E})$ consists of a set of nodes $\mathbf{V} = \{V_1, \ldots, V_n\}$ and a set of edges $\mathbf{E}$. The two ends of an edge are called *marks*. A graph is ***directed mixed*** if the edges in the graph are *directed* ($\rightarrow$), or *bi-directed* ($\leftrightarrow$). A node $V_i$ is a ***parent***, ***child***, or ***spouse*** of a node $V_j$ if there is $V_i \rightarrow V_j$, $V_i \leftarrow V_j$, or $V_i \leftrightarrow V_j$. A ***path*** $\pi$ in $\mathcal{G}$ is a sequence of distinct nodes $\langle V_0, \ldots, V_s \rangle$ such that for $0 \leq i \leq s - 1$, $V_i$ and $V_{i+1}$ are adjacent in $\mathcal{G}$. A ***directed path*** from $V_i$ to $V_j$ is a path composed of directed edges pointing towards $V_j$, and $V_i$ is called an ***ancestor*** of $V_j$ and $V_j$ is a ***descendant*** of $V_i$. An ***almost directed cycle*** happens when $V_i$ is both a spouse and an ancestor of $V_j$. A ***directed cycle*** happens when $V_i$ is both a child and an ancestor of $V_j$.

**MAG and PAG.** A directed mixed graph is ***ancestral*** if it doesn't contain a directed or almost directed cycle. A ***maximal ancestral graph (MAG, denoted by $\mathcal{M}$)*** is an ancestral graph, where for any two non-adjacent nodes, there is a set of nodes that m-separates them. A MAG is a ***directed acyclic graph*** (DAG) if it has only directed edges. The set of MAGs that encode the same m-separation relations forms a ***Markov equivalence class (MEC)***, denoted by $[\mathcal{M}]$, which is uniquely characterized by a ***partial ancestral graph (PAG)***, denoted by $\mathcal{P}$. In a PAG, a tail '$-$' or arrowhead '$>$' occurs if the corresponding mark is tail or arrowhead in all the Markov equivalent MAGs, and a circle '$\circ$' occurs otherwise. For convenience, we use an asterisk (*) to denote any possible mark of a PAG ($\circ, >, -$) or a MAG ($>, -$).

**Possibly relationship and path.** For two vertices $V_i$ and $V_j$ in $\mathcal{P}$, $V_i$ is a ***possibly parent/possibly child/neighbor*** of $V_j$ if there is $V_i \circ\!\!\rightarrow V_j / V_i \leftarrow\!\circ V_j / V_i \circ\!\!-\!\!\circ V_j$ in $\mathcal{P}$. A ***possibly directed path*** from $V_i$ to $V_j$ is a path where every edge without an arrowhead at the mark near $V_i$, and $V_i$ is called a ***possible ancestor*** of $V_j$, and $V_j$ is a ***possible descendant*** of $V_i$. The set of possible descendants of $V_i$ in $\mathcal{G}$ is denoted by $\text{PossDe}(V_i, \mathcal{G})$. A path $\pi$ from $V_i$ to $V_j$ is a ***collider path*** if all the passing nodes are colliders on $\pi$, e.g., $V_i \rightarrow V_{i+1} \leftrightarrow \ldots \leftrightarrow V_{j-1} \leftarrow V_j$.

**Markov Blanket.** Assuming faithfulness, in a MAG, the Markov blanket of a vertex $X$, noted as $\text{MB}(X, \mathcal{M})$, con-

sists of the set of parents, children, children's parents of $X$, as well as the district of $X$ and of the children of $X$, and the parents of each vertex of these districts, where the district of a vertex $V$ is the set of all vertices reachable from $V$ using only bidirected edges.

**Notations.** We use $Adj(V_i)$, $Pa(V_i)$, $De(V_i)$ and $PossDe(V_i)$ to denote the set of *adjacent, parents, descendants* and *possible descendants* of node $V_i$, respectively. We define $Forb(X, Y) = \{W' \in \mathbf{V} \mid W' \in De(W), W \text{ lies}$ on a causal path from $X$ to $Y$ in $\mathcal{G}\}$. We use the notation $\mathbf{X} \perp\!\!\!\perp \mathbf{Y}|\mathbf{Z}$ for "$\mathbf{X}$ is statistically independent of $\mathbf{Y}$ given $\mathbf{Z}$", and $\mathbf{X} \not\perp\!\!\!\perp \mathbf{Y}|\mathbf{Z}$ for the negation of the same sentence (Dawid, 1979). We use w.r.t. to denote "with respect to". $MB^+(X) = \{MB(X) \cup X\}$.

**Definition 1** (**Visible Edges**). *(Zhang, 2008a) Given a MAG $\mathcal{M}$ / PAG $\mathcal{P}$, a directed edge $X \rightarrow Y$ in $\mathcal{M}$ / $\mathcal{P}$ is* visible *if there is a node $S$ not adjacent to $Y$, such that*

- *there is an edge between $S$ and $X$ that is into $X$, or*
- *there is a collider path between $S$ and $X$ that is into $X$ and every non-endpoint node on the path is a parent of $Y$.*

*Otherwise, $X \rightarrow Y$ is said to be* invisible.

**Example 1.** *In Figure 2(c), node $V_3$ can serve as $S$ for the first condition: $V_3$ is not adjacent to $Y$, and the edge between $V_3$ and $X$ is oriented into $X$. So, $X \rightarrow Y$ is visible. In Figure 2(d), node $V_1$ can serve as $S$ for the second condition: there exists a collider path from $V_1$ to $X$ oriented into $X$, and every non-endpoint node on this path is a parent of $Y$. So, $X \rightarrow Y$ is visible.*

**Definition 2** (**Generalized adjustment criterion-GAC**). *(Perković et al., 2018) Let $X$ and $Y$ be two distinct nodes in a MAG or PAG $\mathcal{G}$ over $\mathbf{O}$. Then a set of nodes $\mathbf{Z} \subseteq \mathbf{O} \setminus \{X, Y\}$ satisfies the generalized adjustment criterion relative to $(X, Y)$ in $\mathcal{G}$ if*

- *$\mathcal{G}$ is adjustment amenable relative to $(X, Y)$, which means that the first edge of every possible directed path from $X$ to $Y$ is visible,*
- *$\mathbf{Z} \cap \text{Forb}(X, Y) = \emptyset$, means that $Z$ cannot contain the descendants of any node on any possible directed path from $X$ to $Y$, and*
- *all definite status non-causal paths from $X$ to $Y$ are blocked by $\mathbf{Z}$.*

*If these conditions hold, then the total causal effect of $X$ on $Y$ is identifiable and is given by*

$$f(y \mid \mathrm{do}(x)) = \begin{cases} f(y \mid x) & \text{if } \mathbf{Z} = \emptyset, \\ \int_{\mathbf{z}} f(y \mid x, \mathbf{z}) f(\mathbf{z}) \, d\mathbf{z} & \text{otherwise.} \end{cases} \quad (1)$$

The GAC defined here corresponds to the **valid adjustment set** discussed later in this article; therefore, we will use "valid adjustment set" as the general term throughout.

## 2.2. Problem Definition

We consider a Structural Causal Model (SCM), as described in Pearl (2009), which includes a set of variables $\mathbf{V} = \mathbf{O} \cup \mathbf{U}$, associated with a joint probability distribution $P(\mathbf{V})$. Here, $\mathbf{O}$ represents the set of observed variables, and $\mathbf{U}$ denotes the set of unmeasured latent variables. Each variable $V_i \in \mathbf{V}$ is generated by a function $f_i$ that depends on its parents in the Directed Acyclic Graph (DAG, denoted as $\mathcal{D}$) and an error term $\varepsilon_i$, i.e. , $V_i = f_i(\mathrm{Pa}(V_i, \mathcal{D}), \varepsilon_i)$, where $\mathrm{Pa}(V_i, \mathcal{D})$ denotes the parents of $V_i$ in $\mathcal{D}$. The error terms $\varepsilon_i$ are assumed to be independent of each other.

**Goal.** Given only the observed variables $\mathbf{O}$, where the pretreatment and causal sufficiency assumptions may be violated, we study the problem of estimating the average causal effect of a treatment variable $X \in \mathbf{O}$ on an outcome variable $Y \in \mathbf{O}$. Specifically, we aim to determine whether $X$ has a causal effect on $Y$, and if so, to identify a valid adjustment set $\mathbf{Z}$ for estimating this effect using local learning.

## 3. Local Theory for Covariate Selection

In this section, we investigate the theoretical foundations for estimating the causal effect of $X$ on $Y$ using only local information derived from observational data, even in the presence of latent variables. Since observational data typically determine only a Markov equivalence class of causal graphs rather than a unique causal structure, different graphs in the equivalence class may imply different causal relations between $X$ and $Y$ (Entner et al., 2013). As a result, causal effect estimation falls into three possible cases, each of which is addressed in the following subsections:

**Case 1.** $X$ has a causal effect on $Y$, and the effect is identifiable via a valid adjustment set. (Section 3.1)

**Case 2.** $X$ has no causal effect on $Y$. (Section 3.2)

**Case 3.** $X$ may or may not have a causal effect on $Y$, and the effect is not identifiable from observational data alone. (Section 3.3)

### 3.1. Identifiable Causal Effect (Case 1)

For a target variable pair $(X, Y)$, we first characterize a local boundary for valid adjustment sets using $\mathrm{MB}(X)$.

**Theorem 1.** *Let $\mathbf{O}$ be the set of observed variables, and let $(X, Y)$ be a pair of target variables in $\mathbf{O}$. Then, there exists a subset of $\mathbf{O}$ that is a valid adjustment set for estimating the average causal effect of $X$ on $Y$ if and only if there exists a subset of $\mathrm{MB}(X)$ that is a valid adjustment set for $X$ and $Y$.*

Theorem 1 implies that if there exists an adjustment set relative to $(X, Y)$ within the full observed variable set $\mathbf{O}$,

then there also exists an adjustment set contained entirely in $\mathrm{MB}(X)$. Conversely, if no subset of $\mathrm{MB}(X)$ constitutes a valid adjustment set, then no subset of $\mathbf{O}$ can serve as a valid adjustment set for $(X, Y)$.

**Remark 1.** *In contrast to the local boundary proposed in (Li et al., 2025a), which relies on the* pretreatment assumption *and is defined in terms of* $\mathrm{MB}(Y)$*, our local boundary is characterized by* $\mathrm{MB}(X)$*. In particular,* $\mathrm{MB}(Y)$ *is no longer guaranteed to provide a valid adjustment set; see Example 2 for a counterexample.*

**Example 2.** *Consider the MAG shown in Figure 2(b) and its corresponding PAG in Figure 3, where* $\mathrm{MB}(Y) = \{V_2, V_8\}$ *and* $\mathrm{MB}(X) = \{V_1, V_4, V_5, V_6, V_7\}$*. Both $V_2$ and $V_8$ lie on a causal path from $X$ to $Y$, and therefore cannot be included in any valid adjustment set. In contrast, the set* $\{V_5\} \subset \mathrm{MB}(X)$ *constitutes a valid adjustment set for estimating the total causal effect of $X$ on $Y$. This example demonstrates that when the pretreatment assumption does not hold,* $\mathrm{MB}(Y)$ *is not complete to characterize the local boundary of valid adjustment sets, whereas* $\mathrm{MB}(X)$ *remains complete.*

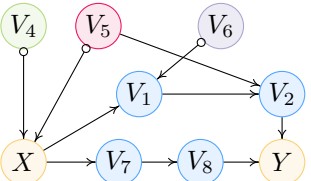

*Figure 3.* The corresponding PAG of the MAG in Figure 2(b)

We now address the problem of locally identifying valid adjustment sets within the local boundary. The identification criteria are formally presented in Theorems 2 and 3.

**Theorem 2 ($\mathcal{R}1$ for Locally Searching Adjustment Sets).** *Let $\mathcal{P}$ be the PAG over $\mathbf{O}$, $(X, Y)$ be a pair of ordered target variables. A subset $\mathbf{Z} \subseteq \mathrm{MB}(X) \setminus \mathrm{PossDe}(X, \mathcal{P})$ is a valid adjustment set w.r.t. $(X, Y)$ if there exists a variable $S \in \mathrm{MB}(X) \setminus (\{Y\} \cup \mathrm{Ch}(X, \mathcal{P}))$ such that (i) $S \not\perp\!\!\!\perp Y \mid \mathbf{Z}$ and (ii) $S \perp\!\!\!\perp Y \mid \mathbf{Z} \cup \{X\}$.*

Condition (i) indicates active paths from $S$ to $Y$ given $\mathbf{Z}$, while condition (ii) ensures no active paths when adding $\mathbf{Z} \cup \{X\}$. Together, these imply that all active paths from $S$ to $Y$ via $\mathbf{Z}$ must pass through $X$, and adding $X$ blocks them. Hence, $\mathbf{Z}$ is valid according to GAC.

Next, we show that $\mathcal{R}1$ of Theorem 2 is not complete for fully identifying valid adjustment sets. This limitation is illustrated by Example 3.

**Example 3.** *Consider the MAG shown in Figure 2(c) and its corresponding PAG in Figure 4. Here,* $\mathrm{Ch}(X, \mathcal{P}) = \{V_2, Y\}$, $\mathrm{MB}(X) = \{V_2, V_3, V_6, Y\}$, *and* $\mathrm{PossDe}(X, \mathcal{P}) = \{V_2, V_5, Y\}$*. However, considering the requirement that $\{S\} \cap \mathbf{Z} = \emptyset$ for conditional independence*

*testing, for any $S \in \{V_3, V_6\}$ and any subset $\mathbf{Z} \subseteq \{V_3, V_6\}$, no combination of $S$ and $\mathbf{Z}$ satisfies the conditions of Theorem 2. Nevertheless, from the given PAG, the set $\{V_3\}$ is a valid adjustment set relative to $(X, Y)$.*

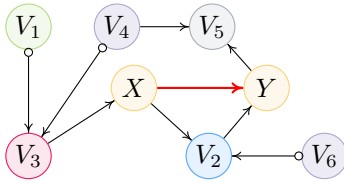

*Figure 4.* The corresponding PAG of the MAG in Figure 2(c)

A closer look at these exceptional cases reveals a common structural property: for every edge adjacent to $X$, the mark at the $X$-endpoint is always determined (i.e., not a circle). This insight directly motivates Theorem 3. Before presenting it, we first introduce the following definition.

**Definition 3.** *Let $\mathcal{P}_{\mathrm{MB}^+(X)}$ be the induced subgraph of $\mathrm{MB}^+(X)$. The definitely non-collider possible parents of $X$ are the nodes that are adjacent to $X$ via an edge of the form $V_0 \circ\!\!\rightarrow X$, where $V_0$ does not act as a collider on any path within this subgraph. This set is denoted as $\mathrm{NCPa}(X, \mathcal{P}_{\mathrm{MB}^+(X)})$.*

**Theorem 3 ($\mathcal{R}2$ for Locally Searching Adjustment Sets).** *Let $\mathcal{P}$ be the PAG over $\mathbf{O}$, $(X, Y)$ be a pair of ordered target variables. Suppose that in the local adjacency structure around $X$, the mark at the $X$-endpoint is always determined. If the causal effect from $X$ to $Y$ is identifiable, then the set $\mathbf{Z} = \mathrm{Pa}(X) \cup \mathrm{NCPa}(X, \mathcal{P}_{\mathrm{MB}^+(X)})$ is a valid adjustment set w.r.t. $(X, Y)$ for identifying the total effect of $X$ on $Y$, and it satisfies $\mathbf{Z} \subseteq \mathrm{MB}(X) \setminus \mathrm{PossDe}(X, \mathcal{P})$.*

Theorem 3 addresses cases where a causal effect from $X$ to $Y$ exists, but Theorem 2 fails to identify a valid adjustment set. In these cases, Theorem 3 provides a valid adjustment set.

**Example 4.** *Now consider Figure 4. When $\mathcal{R}1$ is not applicable, we find that the situation conforming to $\mathcal{R}2$ is that the marks around $X$ are determined, but $S$ and $\mathbf{Z}$ coincide. In this case, we can use Rule 2 to determine a valid adjustment set $\mathbf{Z} = \{V_3\}$.*

### 3.2. No Causal Effect (Case 2)

Next, we consider the scenario where the causal effect is identified as zero. By analyzing local structural features in observational data, we can derive rules that allow us to unambiguously conclude that $X$ has no causal effect on $Y$.

**Theorem 4 ($\mathcal{R}3$ for Locally Identifying No Causal effect).** *Let $\mathcal{P}$ be the PAG over $\mathbf{O}$, $(X, Y)$ be a pair of ordered target variables. Then, $X$ has no causal effect on $Y$ if there exists a subset $\mathbf{Z} \subseteq \mathrm{MB}(X)$ with $\mathbf{Z} \cap \mathrm{PossDe}(X, \mathcal{P}) = \emptyset$ and a variable $S \in \mathrm{MB}(X) \setminus (\{Y\} \cup \mathrm{Ch}(X, \mathcal{P}))$ such that at*

*least one of the following conditions holds: (i) $X \perp\!\!\!\perp Y \mid \mathbf{Z}$, or (ii) $S \not\perp\!\!\!\perp X \mid \mathbf{Z}$ and $S \perp\!\!\!\perp Y \mid \mathbf{Z}$.*

Theorem 4 provides a local criterion for identifying the absence of a causal effect of $X$ on $Y$. Condition (i) implies that conditioning on local variables (excluding descendants of $X$) blocks all active paths between $X$ and $Y$, meaning the total causal effect of $X$ on $Y$ is zero. Condition (ii) captures a complementary structural pattern. In a PAG $\mathcal{P}$, circular marks indicate ambiguous edge orientations, which may give rise to spurious associations between $X$ and $Y$. In this case, there exists a variable $S$ such that $S \not\perp\!\!\!\perp X \mid \mathbf{Z}$ while $S \perp\!\!\!\perp Y \mid \mathbf{Z}$. This implies that, conditional on $\mathbf{Z}$, information can flow from $S$ to $X$ but does not propagate further to $Y$. Consequently, the association between $X$ and $Y$ cannot be explained by a directed causal path from $X$ to $Y$, and thus $X$ has no causal effect on $Y$. We illustrate this scenario with a simple example below.

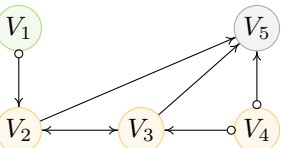

*Figure 5.* The corresponding PAG of the MAG in Figure 2(d)

**Example 5.** *Consider the MAG shown in Figure 2(d) and its corresponding PAG in Figure 5. In Figure 5, consider two possible assignments of variables. **First**, consider the pair $(X, Y) = (V_2, V_4)$. According to the first case of Theorem 4, we observe that $X \perp\!\!\!\perp Y \mid \emptyset$. This marginal independence directly implies that the causal effect of $X$ on $Y$ is zero. **Second**, consider the pair $(X, Y) = (V_2, V_3)$. This pair satisfies the second case of Theorem 4. Specifically, there exists a variable $S = V_1$ that fulfills the following conditions: $S \not\perp\!\!\!\perp X \mid \emptyset$ and $S \perp\!\!\!\perp Y \mid \emptyset$. Consequently, the theorem again allows us to conclude that the causal effect of $X$ on $Y$ is zero.*

### 3.3. Non-Identifiable (Case 3)

We begin by explaining why case 3 is *non-identifiable*. Under standard assumptions, observational data determine only a Markov equivalence class of causal structures, typically represented by a PAG, which encodes all conditional independencies implied by the underlying distribution (Spirtes et al., 2000; Zhang, 2008a; Entner et al., 2013). Within a given equivalence class, different causal structures can agree on all observable (in)dependence relations while differing in whether the directed edge $X \rightarrow Y$ is present. As a result, certain causal relationships cannot be uniquely identified from observational data alone.

**Theorem 5.** *Under causal Markov and Faithfulness assumptions, the causal effect of $X$ on $Y$ must fall into one of the three identifiable cases characterized by Theorem 2,*

*Theorem 3, and Theorem 4. If none of these conditions hold, then the effect is not identifiable from observable conditional independence and dependence relations.*

Specifically, this theorem characterizes the identifiability boundary: if $\mathcal{R}1$ (Theorem 2) or $\mathcal{R}2$ (Theorem 3) applies, the causal relationship is determined as an *Identifiable Non-Zero Effect*, and if $\mathcal{R}3$ (Theorem 4) applies, it is identified as an *Identifiable Zero Effect*. Conversely, if none of the rules $\mathcal{R}1$–$\mathcal{R}3$ apply, then it is impossible to assertain whether $X$ has a causal effect on $Y$ relying solely on the conditional independencies and dependencies among the observed variables; in this case, the effect is *Non-Identifiable* theoretically.

## 4. Practical Algorithm

In this section, we leverage the above local identification rules to propose the **L**ocal **C**ovariate **S**election (LCS) algorithm, which determines whether there exists a causal effect of a variable $X$ on another variable $Y$ and, if so, estimates this effect without bias. Given an ordered variable pair $(X, Y)$, the algorithm consists of the following two key steps:

**Step 1:** Learn the local structure around $X$ using any local learning algorithm.

**Step 2:** Apply the proposed rules to determine a valid adjustment set and estimate the causal effect.

The algorithm uses $\Theta$ to store the estimated causal effect of $X$ on $Y$. If $\Theta$ is null, it corresponds to Case 3 (Non-identifiable effect), indicating that the available observational data is insufficient to determine whether $X$ has a causal effect on $Y$. If $\Theta = 0$, it corresponds to Case 2 (No causal effect), indicating that $X$ has no causal effect on $Y$. Otherwise, a non-zero $\Theta$ corresponds to Case 1 (Non-zero causal effect) and represents the estimated causal effect of $X$ on $Y$. The complete procedure is summarized in Algorithm 1. The local structure learning algorithm used is from (Li et al., 2025b), with details provided in Algorithm 2 in Appendix C.2.

**Theorem 6** (The Soundness and Completeness of Algorithm 1)**.** *Assume oracle conditional independence tests. Under causal Markov and Faithfulness assumptions, the LCS algorithm correctly returns the causal effect $\Theta$ whenever any of the rules $\mathcal{R}1$, $\mathcal{R}2$, or $\mathcal{R}3$ applies. If none of these rules applies, however, then based on the testable conditional independencies and dependencies among the observed variables, the LCS algorithm cannot determine whether $X$ has a causal effect on $Y$.*

Theorem 6 establishes that the LCS algorithm is both *sound* and *complete*. Soundness means that, given an indepen-

---

**Algorithm 1** Local Covariate Selection (LCS)

---

**input** Target variable pair $(X, Y)$, observed data $\mathbf{O}$
1: $G_X \leftarrow$ Selected local structure learning algorithm
2: Obtain $\text{MB}(X), \text{Pa}(X, \mathcal{P}), \text{NCPa}(X, \mathcal{P}_{\text{MB}+(X)})$,
3:        $\text{Ch}(X, \mathcal{P}), \text{PossDe}(X, \mathcal{P}) \leftarrow G_X$
4: $\Theta \leftarrow \emptyset$           // Initialize causal effect estimate
5: **if** $S$ and $\mathbf{Z}$ satisfy $\mathcal{R}1$ (Theorem 2) **then**
6:     $\Theta \leftarrow \theta$           // Case 1: identifiable
7:     **return** $\Theta$
8: **end if**
9: **if** the local structure around $X$ satisfies $\mathcal{R}2$ (Theorem 3) **then**
10:     $\mathbf{Z} \leftarrow \text{Pa}(X) \cup \text{NCPa}(X, \mathcal{P}_{\text{MB}+(X)})$
11:     $\Theta \leftarrow \theta$         // Case 1: identifiable
12:     **return** $\Theta$
13: **end if**
14: **if** $S$ and $\mathbf{Z}$ satisfy $\mathcal{R}3$ (Theorem 4) **then**
15:     **return** $\Theta \leftarrow 0$      // Case 2: zero causal effect
16: **end if**
17: **return** $\Theta$     // Empty $\Theta$ indicates non-identifiable Case 3

---

dence oracle, any inference derived from rules $\mathcal{R}1$, $\mathcal{R}2$, or $\mathcal{R}3$ is guaranteed to be correct whenever the rule is applicable. Completeness, on the other hand, characterizes the identifiability boundary: if none of these rules applies, then it is impossible to determine whether $X$ has a causal effect on $Y$ based solely on the observed conditional independencies and dependencies.

This result is consistent with Theorem 5, which formally shows that when $\mathcal{R}1$–$\mathcal{R}3$ fail to apply, the causal effect is not identifiable from the testable conditional independence and dependence relations among observed variables. Therefore, when LCS cannot determine whether $X$ has a causal effect on $Y$, this should not be interpreted as a limitation of the algorithm itself, but rather as a fundamental limitation of the available observational information: the causal effect cannot be inferred from this class of observable conditional independence/dependence information alone.

**Complexity of the Algorithm.** Let $n = |\mathbf{O}|$ denote the number of observed variables. The computational cost of the proposed algorithm mainly comes from two components. First, the algorithm learns a local causal structure over $\text{MB}^+(X)$. Let $|\text{MB}^+|$ denote the maximum size of the expanded Markov blanket encountered during the procedure. The worst-case complexity of this step is bounded by $\mathcal{O}\left(|\text{MB}^+|^2 2^{|\text{MB}^+|}\right)$. Notably, typically $|\text{MB}^+| \ll n$, then this step can be substantially more efficient than applying a global structure learner to all observed variables. Second, the algorithm applies $\mathcal{R}1$–$\mathcal{R}3$ to verify the existence of a valid adjustment set. This step performs conditional independence tests within the local boundary of $X$. Its complexity is bounded by $\mathcal{O}\left(|\mathbf{S}| 2^{|Z|}\right)$, where $|\mathbf{S}|$ is the size of the search space for auxiliary variables, and $|Z|$ is the size of the search space for constructing candidate adjustment sets. Therefore, the overall complexity of the algorithm is

bounded by $\mathcal{O}\left(|\text{MB}^+|^2 2^{|\text{MB}^+|} + |\mathbf{S}| 2^{|Z|}\right)$.

The worst-case exponential complexity is a direct consequence of the completeness requirement: to ensure no valid adjustment set within the local boundary is overlooked, an exhaustive search over candidate subsets is generally necessary. This challenge is inherent to all complete methods. For instance, the global method EHS exhibits exponential complexity relative to the *entire* set of observed variables. In contrast, local methods like LSAS and LCS significantly reduce this search space. While LSAS is exponential in $|\text{MB}(Y)|$ and relies on the restrictive pretreatment assumption, LCS restricts its exponential search to the local boundary $\text{MB}(X)$ without requiring such an assumption. Consequently, LCS achieves a superior balance between completeness and computational feasibility by localizing the search while maintaining broader applicability.

## 5. Experimental Results

To verify the accuracy and efficiency of our proposed method, we conducted extensive experiments on both synthetic data and on two real-world datasets. We here used the existing implementation of the TC discovery algorithm (Pellet & Elisseeff, 2008b) to find the MB of a target variable. Our source code is included in the supplementary materials.

**Baselines.** We compared our method with the following representative approaches. The EHS algorithm (Entner et al., 2013) performs a global search under the pretreatment assumption without requiring explicit graph learning. The CEELS method (Cheng et al., 2022) conducts a local search under the same pretreatment assumption. The LSAS method (Li et al., 2025b) also operates under the pretreatment assumption, but relaxes the causal sufficiency assumption. In contrast, the LDP method (Maasch et al., 2024) further relaxes the pretreatment assumption in the local search setting.

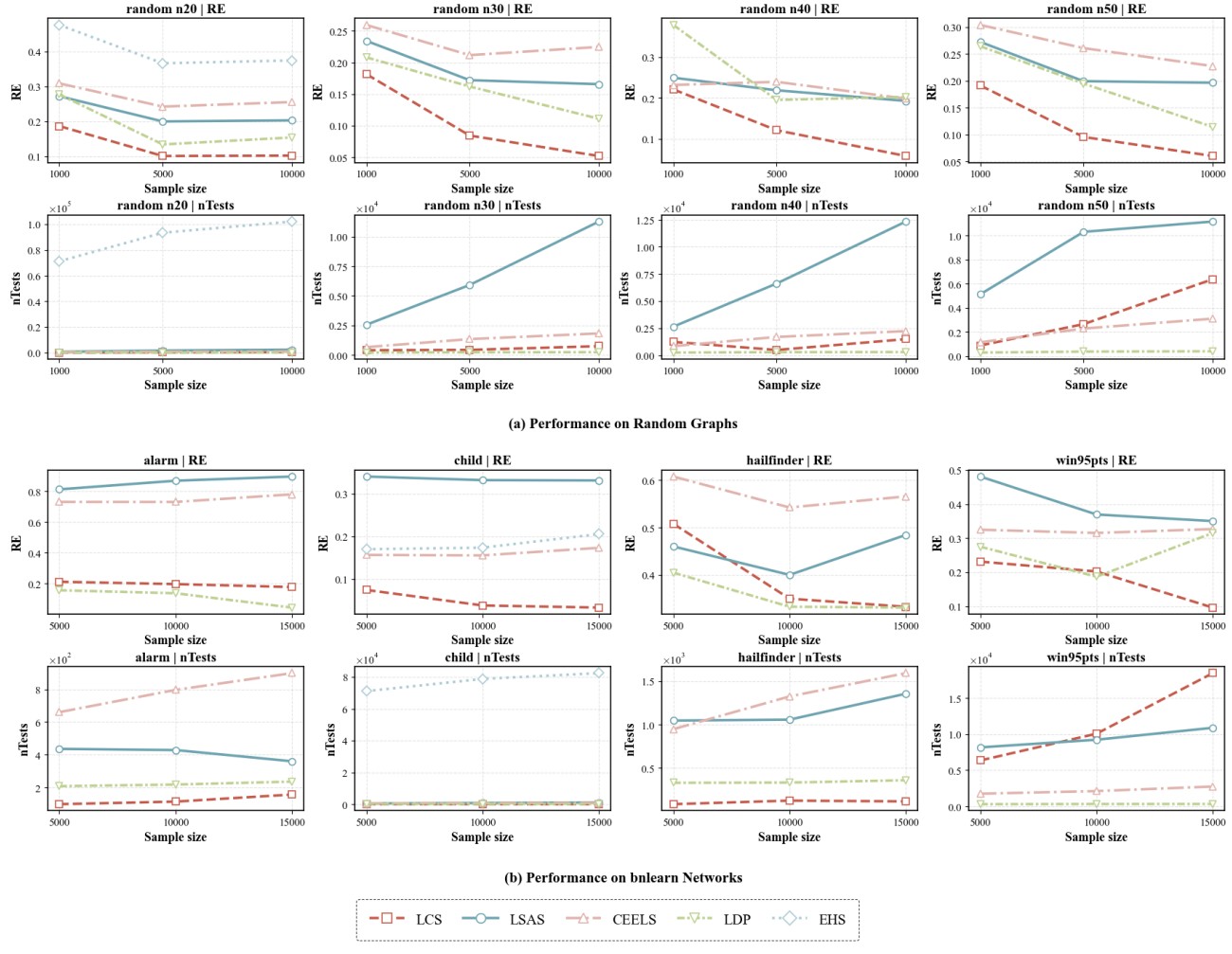

*Figure 6.* Relative error (RE) and the number of conditional independence tests (nTests) across (a) Random Graphs and (b) bnlearn Benchmark Networks. Results are shown as functions of sample size. Lower RE corresponds to higher estimation accuracy, while fewer nTests indicate lower computational cost.

**Evaluation Metrics.** We evaluated the performance of algorithms using the following typical metrics(Cheng et al. (2022), Cheng et al. (2023), Li et al. (2025a)): **Relative Error (RE)** and **nTest**. RE is the relative error (in percentage) between the estimated total causal effect ($\hat{CE}$) and the true total causal effect ($CE$), i.e., $RE = \left| \left( \hat{CE} - CE \right) / CE \right| \times 100\%$. nTest is the number of (conditional) independence tests implemented by an algorithm.

### 5.1. Synthetic Data

**Data Setup.** Following Malinsky & Spirtes (2016); Entner et al. (2013), we generate data from linear Gaussian causal models. Edge coefficients are independently sampled from a Uniform distribution on $[0.5, 1.5]$, and noise terms are drawn from a standard Gaussian distribution. Causal effects are estimated via linear regression, where the effect of $X$ on $Y$ corresponds to the partial regression coefficient of $X$ (Malinsky & Spirtes, 2017). In all experiments, results are reported as the average of 100 independent datasets. For each dataset, latent variables are randomly selected from those nodes that have at least two children, and a target ordered pair $(X, Y)$ is also chosen at random. We performed experiments on two types of graphs, i.e., **Random Graphs** and **Benchmark Networks**. Random graphs are generated using the Erdős-Rényi model $G(n, d)$ (Erdős & Rényi, 1960), with $n \in \{20, 30, 40, 50\}$ and average degree $d = 3$. The number of latent variables is set to $10\% \times n$. Further, we choose four benchmark Bayesian networks: `alarm`, `child`, `hailfinder`, and `win95pts`. These networks contain 37, 20, 56, and 76 nodes, respectively[1]. The number of latent variables is set to 2, 2, 7, and 7, respectively.

#### 5.1.1. PERFORMANCE ON RANDOM GRAPHS

Figure 6 (a) reports the performance on Random Graphs with varying numbers of nodes. Overall, our proposed **LCS** method consistently achieves lower relative error while requiring substantially fewer conditional independence tests across different sample sizes and graph scales, demonstrating a favorable trade-off between estimation accuracy and computational cost. For the EHS method, since the RE and nTests are substantially larger than those of other methods in several settings, we did not display them due to limited space.

*Remark 2. Because LCS includes an initial structure learning stage, the number of conditional independence tests naturally increases as the number of nodes grows, which is an expected consequence of the algorithmic design.*

[1]Additional information is available at https://www.bnlearn.com/bnrepository/.

#### 5.1.2. PERFORMANCE ON BENCHMARK NETWORKS

Figure 6 (b) reports the estimation accuracy and computational cost across different Benchmark Networks.

Regarding **Relative Error (RE)**, **LCS** consistently achieves the lowest or near-lowest error across all networks and sample sizes. This advantage is particularly pronounced on larger and more complex networks, such as `win95pts`, where LCS maintains stable and low RE as the sample size increases. In contrast, LSAS and CEELS exhibit substantially higher errors in these settings, primarily due to the violation of pretreatment assumptions. While LDP benefits from increased sample sizes, it remains notably inferior to LCS and is excluded from the plot as its performance on the `child` dataset exceeds the current scale.

In terms of **nTests (Computational Cost)**, LCS requires a higher number of tests on the `win95pts` network compared to simpler benchmarks. This is expected, as the substantial node count in `win95pts` necessitates extensive structure learning, naturally leading to an increased number of conditional independence tests. When applicable, EHS shows even higher computational costs alongside unstable performance. Moreover, due to the extremely large number of nTests required by EHS, its results are partially truncated or not fully visible in several panels.

### 5.2. Performance on Real-World Datasets

#### 5.2.1. CATTANEO2 DATASET

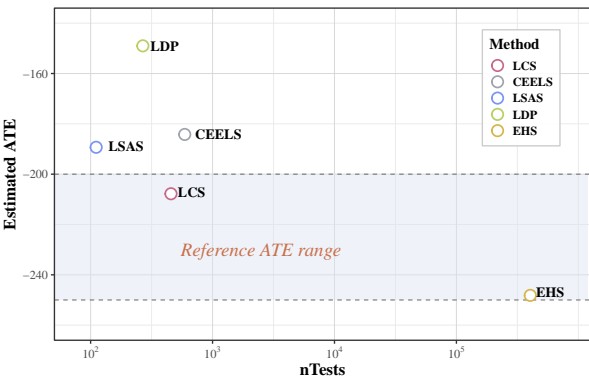

*Figure 7.* Estimation results based on Cattaneo2 dataset

We evaluate our proposed method on a real-world dataset, the Cattaneo2 dataset, which contains birth outcomes for 4,642 singleton deliveries in Pennsylvania, USA (Cattaneo, 2010; Almond et al., 2005). Our goal is to estimate the causal effect of maternal smoking during pregnancy ($X$) on infant birth weight ($Y$). Starting from the 23 covariates commonly used in prior studies, we intentionally extend the feature space by adding one descendant variable of the treatment, resulting in a dataset with 24 observed

variables. This design deliberately violates the pretreatment assumption, creating a more challenging and realistic setting for causal effect estimation. The covariates capture key maternal and family characteristics, including demographic, educational, and health-related factors. The dataset is publicly available from the National Center for Health Statistics (NCHS) (`https://ftp.cdc.gov/pub/Health_Statistics/NCHS/Datasets/DVS/natality/`).

**Reference effect.** Since neither the true causal graph nor the ground-truth causal effect is known for this dataset, we follow the empirical consensus established in Almond et al. (2005). Using regression-adjusted and propensity score–based estimators, they report a substantial negative effect of maternal smoking on birth weight, typically ranging between 200 and 250 grams. We therefore treat this interval as a reasonable reference range for assessing the plausibility of estimated effects.

**Result.** Figure 7 presents a joint comparison of estimation accuracy and computational complexity on real-world data. When potential descendants of the treatment are included among the observed variables, substantial differences arise across methods. Notably, **LCS** most closely approximates the reference effect while requiring a relatively small number of conditional independence tests.

In contrast, CEELS, LSAS, LDP, and EHS either deviate more from the reference effect or incur substantially higher computational costs. Overall, these results demonstrate the robustness of **LCS** under violations of the pretreatment assumption. By explicitly leveraging local causal structure and avoiding inappropriate adjustment for post-treatment variables, **LCS** achieves a favorable balance between estimation accuracy and computational efficiency.

### 5.2.2. JOBS DATASET

We conduct experiments on the widely used Jobs dataset, which is constructed based on the original LaLonde experimental study (LaLonde, 1986) and augmented with observational control samples from the Panel Study of Income Dynamics (PSID), following the benchmark setup in Imai & Ratkovic (2014). Specifically, the dataset contains 2,936 samples with 10 variables and is commonly used to study the causal effect of a job training program on earnings for treated individuals.

**Reference effect.** Following Imai & Ratkovic (2014) and Cheng et al. (2020), the ground-truth Average Treatment Effect on the Treated (ATT) is taken as $886, which is derived from the randomized experimental component of the LaLonde study. This value serves as the reference for evaluating estimation accuracy. We report the relative bias

*Table 2.* Results on the Jobs dataset.

| Method | ATT | Bias (%) | nTests |
|---|---|---|---|
| EHS | 930.606 | 5.04 | 1732 |
| LSAS | 989.346 | 11.67 | 176 |
| CEELS | 683.406 | 22.87 | 183 |
| LDP | -13597.588 | 1635.98 | 61 |
| **LCS (Ours)** | **854.528** | **3.55** | 303 |

defined as

$$\text{Bias}(\%) = \frac{|\widehat{\text{ATT}} - \text{ATT}^*|}{|\text{ATT}^*|} \times 100,$$

where $\text{ATT}^* = \$886$ is the reference effect.

**Result.** As shown in Table 2, the proposed LCS method achieves the lowest estimation bias among all compared methods, indicating that it produces treatment effect estimates closest to the benchmark ATT. Compared with the global method EHS, LCS not only improves estimation accuracy but also requires significantly fewer conditional independence tests, demonstrating better computational efficiency.

In comparison with local methods such as LSAS, CEELS, and LDP, LCS consistently yields more accurate estimates. Notably, LDP exhibits extremely large bias due to unstable adjustment set selection, highlighting the importance of reliable local structure learning. Overall, these results demonstrate that LCS achieves a favorable balance between accuracy and efficiency, making it well-suited for causal effect estimation in real-world observational settings with potential hidden confounders.

## 6. Conclusion

In this paper, we propose a novel method for locally selecting adjustment sets relative to a target causal relationship. Our method is both sound and complete, without relying on the assumptions of causal sufficiency or pretreatment. Specifically, we characterize the local existence boundary of valid adjustment sets and develop a set of theoretical tools to identify such sets within this boundary. Building upon these theoretical results, we design a fully local algorithm capable of selecting a valid adjustment set for estimating the unbiased total effect of a treatment on an outcome. We evaluate the proposed method through extensive experiments on both synthetic and real-world datasets, demonstrating its outperformance in terms of both accuracy and computational efficiency compared to existing approaches. In future work, incorporating expert knowledge into the adjustment set selection process could be explored to further enhance the performance of our method (Perkovic et al., 2017; Fang & He, 2020; Wang et al., 2023b).

# Acknowledgements

We appreciate the comments from anonymous reviewers, which greatly helped to improve the paper. This research was supported by the National Natural Science Foundation of China (62306019, 62472415). YZ would like to acknowledge the support of Beijing Natural Science Foundation (4264116), Research Foundation for Youth Scholars of Beijing Technology and Business University (RFYS2025), and General Project of Science and Technology Program of Beijing Municipal Education Commission (KM202410011016). FX was supported by the China Scholarship Council (CSC), the Beijing Key Laboratory of Applied Statistics and Digital Regulation, and the BTBU Digital Business Platform Project by BMEC. KZ would like to acknowledge the support from NSF Award No. 2229881, AI Institute for Societal Decision Making (AI-SDM), the National Institutes of Health (NIH) under Contract R01HL159805, and grants from Quris AI, Florin Court Capital, MBZUAI-WIS Joint Program, and the Al Deira Causal Education project.

# Impact Statement

This paper presents work whose goal is to advance the field of Causal Inference. There are many potential societal consequences of our work, none which we feel must be specifically highlighted here.

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

## Appendix Contents

# A. More Details on Preliminaries

## A.1. Notation list

*Table 3.* The list of main notations used in this paper

| Notation | Description |
|---|---|
| $G$ | A directed acyclic graph (DAG) |
| $\mathcal{M}$ | A Maximal Ancestral Graph (MAG) |
| $\mathcal{P}$ | A Partial Ancestral Graph (PAG) |
| $[\mathcal{M}]$ | A class of Markov equivalent MAGs |
| $[\mathcal{P}]$ | The Markov equivalence class represented by PAG |
| $\mathbf{V}$ | The set of all variables |
| $\mathbf{O}$ | The set of observed variables |
| $\mathbf{L}$ | The set of latent variables |
| $\mathrm{MB}(X, \mathcal{M})$ | The Markov blanket of a vertex $X$ in a MAG $\mathcal{M}$ |
| $\mathrm{MB}(X, \mathcal{P}), \mathrm{MB}(X)$ | The Markov blanket of a vertex $X$ in a PAG $\mathcal{P}$, and we use $\mathrm{MB}(X)$ to denote $\mathrm{MB}(X, \mathcal{P})$ when there is no loss in clarity |
| $\mathrm{MB}^+(X)$ | The set of $\{\mathrm{MB}(X) \cup X\}$ |
| $\mathrm{Pa}(X, \mathcal{P}), \mathrm{Ch}(X, \mathcal{P})$ | The set of all parents ($\to X$) and children ($\leftarrow X$) of $X$ in a PAG $\mathcal{P}$, respectively |
| $\mathrm{Adj}(X, \mathcal{G})$ | The set of all variables that have direct edges connected to $X$ |
| $\mathrm{NCPa}(X, \mathcal{P}_{\mathrm{MB}+(X)})$ | The set of possible parents of $X$ that cannot act as colliders in $\mathcal{P}_{\mathrm{MB}+(X)}$, as defined in Definition 3. |
| $\mathrm{Pa}^*(X, \mathcal{P})$ | The augmented parent set of $X$ in a PAG $\mathcal{P}$ |
| $\mathrm{An}(X, \mathcal{M}), \mathrm{De}(X, \mathcal{M})$ | The set of all ancestors and descendants of $X$ in a MAG $\mathcal{M}$, respectively |
| $\mathrm{Dis}(X, \mathcal{M})$ | The district of $X$ |
| $\mathrm{Forb}(X, Y)$ | The set of all possible descendants of $W$, and $W$ lies on a possibly directed path from $X$ to $Y$ in $\mathcal{G}$ |
| $\mathbf{X} \perp\!\!\!\perp \mathbf{Y} \mid \mathbf{Z}$ | $\mathbf{X}$ is statistically independent of $\mathbf{Y}$ given $\mathbf{Z}$. |
| $\mathbf{X} \not\perp\!\!\!\perp \mathbf{Y} \mid \mathbf{Z}$ | $\mathbf{X}$ is not statistically independent of $\mathbf{Y}$ given $\mathbf{Z}$ |
| $R_{\underline{X}} = R_{\underline{X}}(\mathcal{R}, \mathcal{G}, X)$ | The graph obtained from $\mathcal{R}$ by removing all directed edges out of $X$ that are visible in $\mathcal{G}$ |
| $\mathcal{G}_{\underline{X}}$ | The graph obtained from $\mathcal{G}$ by removing all visible directed edges out of $X$ in $\mathcal{G}$ |

## A.2. Detailed definitions about the graph

**Definition 4** (**m-connecting** (Spirtes et al., 2000; Zhang, 2008b))**.** *In a directed mixed graph, a path $\pi$ between vertices $X$ and $Y$ is **m-connecting (acitve)** relative to a (possibly empty) set of vertices $\mathbf{Z}$ ($X, Y \notin \mathbf{Z}$) if (1) every non-collider on $\pi$ is not a member of $\mathbf{Z}$; (2) every collider on $\pi$ has a descendant in $\mathbf{Z}$.*

$\mathbf{X}$ and $\mathbf{Y}$ are said to be **m-separated** by $\mathbf{Z}$ if there is no active path between any vertex in $\mathbf{X}$ and any vertex in $\mathbf{Y}$ relative to $\mathbf{Z}$.

**Definition 5** (**Inducing Path**)**.** *An **inducing path** between $V_i$ and $V_j$ is a path on which every non-endpoint vertex is a collider on the path and every collider is an ancestor of either $V_i$ or $V_j$.*

**Definition 6** (**Maximal Ancestral Graph** (Richardson & Spirtes, 2002; Zhang, 2008b))**.** *A directed mixed graph is **ancestral** if it doesn't contain a directed or almost directed cycle. An ancestral graph is called **maximal** if for any two non-adjacent vertices, there is no inducing path between them, i.e. , there exists a set of vertices that m-separates them. A directed mixed graph is called a* maximal ancestral graph *(**MAG**) if it is ancestral and maximal, denoted by $\mathcal{M}$.*

A MAG is a *Directed Acyclic Graph* (DAG) if it has only directed edges. Two MAGs are ***Markov equivalent*** if they share the same m-separations.

**Definition 7** (**Partial Ancestral Graph** (Spirtes et al., 2000; Zhang, 2006))**.** *Let $[\mathcal{M}]$ be the Markov equivalence class of an underlying MAG $\mathcal{M}$. A partial ancestral graph (**PAG, denoted by $\mathcal{P}$**) represents the equivalence class $[\mathcal{M}]$, where a tail '−' or arrowhead '>' occurs if the corresponding mark is tail or arrowhead in every $\mathcal{M} \in [\mathcal{M}]$, and a circle '∘' occurs otherwise.*

**Definition 8** (**Markov Equivalence**)**.** *Two MAGs $\mathcal{M}_1$ and $\mathcal{M}_2$ are Markov equivalence if they share the same m-separations.*

Basically a Partial Ancestral Graph represents an equivalence class of MAGs.

**Definition 9** (**Causal Markov Condition** (Spirtes et al., 2000)). *Given a set of variables $\mathbf{V}$ whose causal structure is represented by a DAG $G$, every variable in $\mathbf{V}$ is probabilistically independent of its non-descendants in $G$ given its parents in $G$.*

**Definition 10** (**Causal Faithfulness Condition** (Spirtes et al., 2000)). *Given a set of variables $\mathbf{V}$ whose causal structure is represented by a DAG $G$, the joint probability of $\mathbf{V}$, $P(\mathbf{V})$, is faithful to $G$ in the sense that $P(\mathbf{V})$ implies no conditional independence relations not already entailed by the causal Markov condition.*

Under the above two conditions, conditional independence relations among the observed variables correspond exactly to m-separation in the MAG or PAG $\mathcal{G}$, i.e., $(\mathbf{X} \perp\!\!\!\perp \mathbf{Y}|\mathbf{Z})_P \Leftrightarrow (\mathbf{X} \perp\!\!\!\perp \mathbf{Y}|\mathbf{Z})_{\mathcal{G}}$.

**Definition 11** (**Markov Blanket**). *The Markov blanket of a variable $X$ is the smallest set of variables $MB(X)$ such that for all $V \in \mathbf{V} \setminus (MB(X) \cup \{X\}$, $X$ is conditionally independent of $V$ given $MB(X)$:*

$$X \perp\!\!\!\perp V \mid MB(X).$$

**Definition 12** (**Markov Blanket for DAGs** (Pearl, 1988; 2000)). *Assuming faithfulness, in a DAG, the Markov blanket of a vertex $X$ is unique, denoted as $MB(X, G)$, includes the set of parents, children, and the parents of the children (spouses) of $X$.*

**Definition 13** (**District**). *In a MAG $\mathcal{M}$, the pulsed district of $X$, denoted as $Dis^+(X, \mathcal{M})$, is the set of vertices connected to $X$ by a path consisting entirely of bidirected edges ($\leftrightarrow$), including $X$ itself. Formally:*

$$Dis^+(X, \mathcal{M}) = \{X\} \cup \{V \mid X \leftrightarrow \cdots \leftrightarrow V \text{ in } \mathcal{M}\}.$$

*The district of $X$ is defined as:*

$$Dis(X, \mathcal{M}) = Dis^+(X, \mathcal{M}) \setminus \{X\}.$$

**Definition 14** (**Markov Blanket for MAGs** (Richardson, 2003; Pellet & Elisseeff, 2008a)). *Assuming faithfulness, in a MAG, the Markov blanket of a vertex $X$, noted as $MB(X, \mathcal{M})$, consists of the set of parents, children, children's parents of $X$, as well as the district of $X$ and of the children of $X$, and the parents of each vertex of these districts, where the district of a vertex $V$ is the set of all vertices reachable from $V$ using only bidirected edges.*

**Definition 15** (**Amenability**). *Let $X$ and $Y$ be two distinct nodes in a MAG or PAG. The MAG or PAG is said to be adjustment amenable relative to $(X, Y)$ if all possibly directed paths from $X$ to $Y$ start with a visible directed edge out of $X$.*

**Definition 16** (**Forbidden set**). *Let $X$ and $Y$ be two distinct nodes in a MAG or PAG $\mathcal{G}$ over $\mathbf{O}$. Then the forbidden set relative to $(X, Y)$ is defined as $\text{Forb}(X, Y) = \{W' \in \mathbf{O} \mid W' \in \text{PossDe}(W), W \text{ lies on a possibly directed path from } X \text{ to } Y \text{ in } \mathcal{G}\}$.*

**Definition 17** (**Arrow-Collider Path**(Li et al., 2025b)). *In a PAG or a MAG, a path $\pi = \langle V_0, \ldots, V_n \rangle$ is called an arrow-collider path from $V_0$ to $V_n$ if every non-endpoint vertex is a collider on $\pi$, and the edge between $V_0$ and $V_1$ points into $V_0$, i.e., $V_0 \leftrightarrow V_1 \cdots \leftarrow\!\ast V_n$. If $n = 1$, $\pi$ simplifies to $V_0 \leftarrow\!\ast V_1$.*

**Definition 18** (**Augmented Parent Set**(Li et al., 2025b)). *Let $\mathcal{G}$ be a PAG or a MAG. The augmented parent set of a vertex $X$, denoted as $Pa^*(X, \mathcal{G})$, is defined as follows: for any vertex $V \in \mathbf{O}$, $V \in Pa^*(X, \mathcal{G})$ if and only if there exists an arrow-collider path $\pi$ from $X$ to $V$ such that:*

*(1) in a MAG, $X$ is a non-ancestor of every vertex on $\pi$, including $V$.*
*(2) in a PAG, $X$ is an invariant non-ancestor of all vertices on $\pi$, including $V$.*

# B. Proofs

### B.1. Proof of Theorem 1

Before presenting the proof, we quote the following Lemma since it is used to prove Theorem 1.

**Lemma 1.** *[Theorem 1 of Xie et al. (2024)] Let $Y$ be any node in $\mathbf{O}$, and $X$ be a node in $MB(Y)$. Then $Y$ and $X$ are m-separated by a subset of $\mathbf{O} \setminus \{Y, X\}$ if and only if they are m-separated by a subset of $MB(Y) \setminus \{X\}$.*

*Proof.* According to Definition 2, a set $\mathbf{Z}$ is a valid adjustment set with respect to $(X, Y)$ in $\mathcal{P}$ if it satisfies all the conditions specified therein. When the treatment variable $X$ is a singleton, the generalized adjustment criterion becomes equivalent to the *generalized back-door criterion* proposed by Maathuis & Colombo (2015). Under our problem setting, where we do *not* impose the pretreatment assumption (i.e., variables in $\mathbf{O}$ are not required to be non-descendants of $X$ and $Y$), the above conditions can be simplified as follows:

A set $\mathbf{Z} \subseteq \mathbf{O}$ is a valid adjustment set for $(X, Y)$ if:

1. the graph $\mathcal{P}$ is *adjustment amenable* relative to $(X, Y)$, meaning that every possible causal path from $X$ to $Y$ begins with a visible directed edge out of $X$;

2. $\mathbf{Z} \cap \mathrm{Forb}(X, Y) = \emptyset$; and

3. $\mathbf{Z}$ $m$-separates $X$ and $Y$ in $\mathcal{P}_{\underline{X}}$.

Here, $\mathcal{P}_{\underline{X}}$ denotes the graph obtained from $\mathcal{P}$ by removing all visible directed edges out of $X$ (Maathuis & Colombo, 2015). Because all such outgoing edges are removed, $X$ has no children in $\mathcal{P}_{\underline{X}}$, and thus no element of $\mathrm{Forb}(X, Y)$ can belong to the Markov blanket of $X$ in $\mathcal{P}_{\underline{X}}$. Consequently, for any candidate separator contained in $\mathrm{MB}(X)$, the condition $\mathbf{Z} \cap \mathrm{Forb}(X, Y) = \emptyset$ is automatically satisfied in $\mathcal{P}_{\underline{X}}$. Therefore, in $\mathcal{P}_{\underline{X}}$, the criterion further simplifies to: For any $\mathbf{Z}' \subseteq \mathbf{O}$, $\mathbf{Z}'$ is a valid adjustment set if and only if it $m$-separates $X$ and $Y$ in $\mathcal{P}_{\underline{X}}$. This simplified form will be used in the proof of our locality theorem. Recall that the theorem states: A subset of $\mathbf{O}$ is a valid adjustment set for estimating the average causal effect of $X$ on $Y$ if and only if there exists a subset of $MB(X)$ that is a valid adjustment set for $X$ and $Y$.

Using the characterization above, this is equivalent to the following purely graph-theoretic statement in $\mathcal{P}_{\underline{X}}$: There exists a subset $\mathbf{Z} \subseteq \mathbf{O}$ that is a valid adjustment set for $(X, Y)$ if and only if there exists a subset $\mathbf{Z} \subseteq \mathrm{MB}(X)$ that is a valid adjustment set for $(X, Y)$. Since, in $\mathcal{P}_{\underline{X}}$, a set is a valid adjustment set if and only if it $m$-separates $X$ and $Y$, the theorem can be restated as:

$$\exists \mathbf{Z} \subseteq \mathbf{O} \text{ s.t. } Z \text{ m-separates } X \text{ and } Y \text{ in } \mathcal{P}_{\underline{X}} \quad \Leftrightarrow \quad \exists \mathbf{Z} \subseteq \mathrm{MB}(X) \text{ s.t. } Z \text{ } m\text{-separates } X \text{ and } Y \text{ in } \mathcal{P}_{\underline{X}}.$$

Note that $\mathrm{MB}'(X) \subseteq \mathrm{MB}(X)$, and $Y$ may not belong to $\mathrm{MB}'(Y)$ in $\mathcal{P}_{\underline{X}}$. We now analyze two cases:

**Case 1:** Suppose that $\mathcal{P}$ is adjustment amenable relative to $(X, Y)$, and that $Y \in \mathrm{MB}(X)$ in $\mathcal{P}_{\underline{X}}$. If $Y \in \mathrm{Adj}(X)$ in $\mathcal{P}_{\underline{X}}$, this contradicts the assumption of adjustment amenability relative to $(X, Y)$. Hence, we must have $Y \notin \mathrm{Adj}(X)$. In this case, according to Lemma 1, there must exist a set $\mathbf{Z}$ that $m$-separates $X$ and $Y$ in $\mathcal{P}_{\underline{X}}$.

**Case 2:** Suppose that $\mathcal{P}$ is adjustment amenable relative to $(X, Y)$, and that $Y \notin \mathrm{MB}(X)$ in $\mathcal{P}_{\underline{X}}$. It then follows that $Y \notin Adj(X)$. Thus, $X$ and $Y$ are $m$-separated by $\mathrm{MB}(X)$, i.e., $(X \perp\!\!\!\perp Y \mid \mathrm{MB}(X))_{\mathcal{P}_{\underline{X}}}$. If, instead, $(X \not\perp\!\!\!\perp Y \mid \mathrm{MB}(X))_{\mathcal{P}_{\underline{X}}}$ were to hold, this would contradict the assumption that $\mathcal{P}$ is adjustment amenable relative to $(X, Y)$. Consequently, no subset of $\mathbf{O} \setminus \{X\}$ is a valid adjustment set w.r.t. $(X, Y)$ in $\mathcal{P}$. $\square$

### B.2. Proof of Theorem 2

*Proof.* Suppose there exists a variable $S \in \mathrm{MB}(X) \setminus (\{Y\} \cup \mathrm{Ch}(X, \mathcal{P}))$ and an adjustment set $\mathbf{Z} \subseteq \mathrm{MB}(X)$ with $\mathbf{Z} \cap \mathrm{PossDe}(X, \mathcal{P}) = \emptyset$ such that (i) $S \not\perp\!\!\!\perp Y \mid \mathbf{Z}$ and (ii) $S \perp\!\!\!\perp Y \mid \mathbf{Z} \cup \{X\}$. We show that under these conditions $\mathbf{Z}$ blocks all non-causal paths between $X$ and $Y$, making it a valid adjustment set.

*Step 1: Why every active path from $S$ to $Y$ must pass through $X$.* From $(S \not\perp\!\!\!\perp Y \mid \mathbf{Z})$, there exists at least one active path from $S$ to $Y$ given $\mathbf{Z}$. From $(S \perp\!\!\!\perp Y \mid \mathbf{Z} \cup \{X\})$, all such paths are blocked once $X$ is added to the conditioning set. Therefore, every originally active path from $S$ to $Y$ must pass through $X$. By construction of $S$ and $\mathbf{Z}$, we also have $\mathbf{Z} \subseteq \mathrm{MB}(X)$ and $\mathbf{Z} \cap \mathrm{PossDe}(X, \mathcal{P}) = \emptyset$. Thus, none of the descendants of $X$ are conditioned on. If $X$ were a collider along an $S$ to $Y$ path, conditioning on $\mathbf{Z}$ would not activate that path, but conditioning on $\mathbf{Z} \cup \{X\}$ would activate it. Therefore, $X$ can't be a collider.

*Step 2: Existence of an active $S* \to X$ path.* From $(S \not\perp\!\!\!\perp Y \mid \mathbf{Z})$ and $(S \perp\!\!\!\perp Y \mid \mathbf{Z} \cup \{X\})$, it follows that, given $\mathbf{Z}$, there must exist at least one active path from $S$ to $X$. Due to the selection scope of $S$, $X$ cannot be a collider. Therefore, the only path between $S$ and $X$ is $S* \to X \to \dots$

*Step 3: No active non-causal path between $X$ and $Y$ can remain.* Assume, toward a contradiction, that there exists an active non-causal path between $X$ and $Y$ given $\mathbf{Z}$. By concatenating this path with the active $S* \to X$ segment and applying Lemma 3.3.1 of Spirtes et al. (2000) (activated when two paths share more than one node), the collider at $X$ yields an active $S \to Y$ path under $\mathbf{Z} \cup \{X\}$. This implies $S \not\perp\!\!\!\perp Y \mid \mathbf{Z} \cup \{X\}$, which contradicts condition (ii). Hence, no such activated non-causal path can exist. Under Condition (ii), $\mathbf{Z}$ blocks every non-causal path from $X$ to $Y$ while retaining all directed causal paths. Therefore, $\mathbf{Z}$ satisfies the generalized adjustment criterion and is a valid adjustment set for $(X, Y)$. $\qquad\square$

## B.3. Proof of Theorem 3

*Proof.* In this case, every neighbor of $X$ is either a definite parent $(\text{Pa}(X))$, a definite non-collider possible parent $(\text{NCPa}(X, \mathcal{P}_{\text{MB}^+(X)}))$, or a definite child $(\text{Ch}(X))$. By the local Markov property, $X$ is conditionally independent of all its non-descendants given its parent set $\text{Pa}(X)$. The graph $\mathcal{P}_{\text{MB}^+(X)}$ represents a Markov equivalence class that enumerates all possible MAGs consistent with the learned local structure around $X$. Let $\mathbf{Z} = \text{Pa}(X) \cup \text{NCPa}(X, \mathcal{P}_{\text{MB}^+(X)})$, which is the union of the parent sets of $X$ across all MAGs in this equivalence class. In other words, in any MAG belonging to this equivalence class, the parent nodes of $X$ are contained in $\mathbf{Z}$.

If $X$ and $Y$ are not conditionally independent given $\text{Pa}(X)$, then any active path from $X$ to $Y$ must be a possibly directed causal path beginning with an edge of the form $\text{Pa}(X) \to X$ followed by an edge $X \to *$. Moreover, every non-causal path between $X$ and $Y$ must pass through a parent of $X$, since all colliders and edge orientations incident to $X$ are fixed. Hence, the set $\text{Pa}(X) \cup \text{NCPa}(X, \mathcal{P}_{\text{MB}^+(X)})$ satisfies the generalized adjustment criterion (Maathuis & Colombo, 2015) and constitutes a valid adjustment set for identifying the total causal effect of $X$ on $Y$. $\qquad\square$

## B.4. Proof of Theorem 4

*Proof.* For case (i), suppose that the causal effect from $X$ to $Y$ is non-zero. By the construction of $\mathbf{Z}$, the set $\mathbf{Z}$ does not contain any descendants of $X$. Moreover, by the Markov blanket property, $\mathbf{Z}$ blocks all non-causal paths between $X$ and $Y$. Thus, any active path from $X$ to $Y$ given $\mathbf{Z}$ must be a directed causal path from $X$ to $Y$. In particular, if the effect is non-zero, then $Y$ is a descendant of $X$ and, given $\mathbf{Z}$, we must have $X \not\perp\!\!\!\perp Y \mid \mathbf{Z}$, which contradicts the condition $X \perp\!\!\!\perp Y \mid \mathbf{Z}$.

For case (ii), the condition $S \not\perp\!\!\!\perp X \mid \mathbf{Z}$ implies that, given $\mathbf{Z}$, there exists an active path $\pi$ from $S$ to $X$ whose last edge points into $X$, and, by construction, neither $S$ nor the nodes in $\mathbf{Z}$ are descendants of $X$. Now suppose that there exists a directed path from $X$ to $Y$. Then, by appending this directed path to $\pi$, we obtain an active path from $S$ to $Y$ of the form $S* \to \cdots \to X \to \cdots \to Y$, which remains active given the set $\mathbf{Z}$, because $X \notin \mathbf{Z}$. This would imply $S \not\perp\!\!\!\perp Y \mid \mathbf{Z}$, contradicting the condition $S \perp\!\!\!\perp Y \mid \mathbf{Z}$. Therefore, it is impossible for a directed path to exist from $X$ to $Y$ in this case. $\quad\square$

## B.5. Proof of Theorem 5

*Proof.* We begin by recalling the *generalized adjustment criterion* (GAC). A set of nodes $\mathbf{Z} \subseteq V \setminus \{X, Y\}$ satisfies the GAC relative to $(X, Y)$ in a graph $\mathcal{G}$ if:

(1) $\mathcal{G}$ is adjustment amenable relative to $(X, Y)$;

(2) $\mathbf{Z} \cap Forb(X, Y) = \emptyset$;

(3) all definite-status non-causal paths from $X$ to $Y$ are blocked by $\mathbf{Z}$.

In $\mathcal{R}1$, $\mathcal{R}1$ and $\mathcal{R}3$, we restrict attention to auxiliary variables $S \in \text{MB}(X) \setminus (\{Y\} \cup \text{Ch}(X, \mathcal{P}))$, and adjustment sets $Z \subseteq \text{MB}(X)$ with $\mathbf{Z} \cap \text{PossDe}(X, \mathcal{P}) = \emptyset$. Under these restrictions, Condition (2) of the GAC is automatically satisfied. Therefore, it suffices to consider only Conditions (1) (amenability) and (3) (blocking of non-causal paths). In particular, we analyze the following two cases, depending on whether a causal effect exists between $X$ and $Y$.

**Case 1: There exists a causal effect from $X$ to $Y$.** Since a causal effect from $X$ to $Y$ can only exist if the first edge on any possibly directed path from $X$ to $Y$ is visible, we distinguish two sub-cases depending on the role played by an auxiliary variable $S$ in determining the visibility of this edge.

**(i) $S$ is not on any non-causal path from $X$ to $Y$.** In this situation, any association between $S$ and $Y$ must be mediated through $X$. Adjustment amenability of $\mathcal{G}$ relative to $(X, Y)$ requires that the first edge out of $X$ on every causal path from $X$ to $Y$ be *visible*. Two visibility patterns are possible.

*Visible case 1:* $S * \to X \to C_1 \to Y$. Since $S \in \mathrm{MB}(X) \setminus (\{Y\} \cup \mathrm{Ch}(X, \mathcal{P}))$ and $S$ is not on any non-causal path from $X$ to $Y$, $\mathcal{R}1$ must hold: (i) $S \not\perp\!\!\!\perp Y \mid \mathbf{Z}$ and (ii) $S \perp\!\!\!\perp Y \mid \mathbf{Z} \cup \{X\}$.

*Visible case 2:* There exists a path from $V$ to $C_1$ ($V * \leftrightarrow\leftrightarrow ... \leftrightarrow X * - * C_1$), the path from $V$ into $X$ is a collider path such that every non-endpoint node on the path is a parent of $C_1$. Under this visibility pattern, an analogous argument applies, and $\mathcal{R}1$ again holds.

In summary, as long as the edge pointed to by $X$ is visible, that is, as long as the first criterion of GAC is satisfied, our rule will definitely be able to identify it.

**(ii) Every possible $S$ lies on a non-causal path from $X$ to $Y$.** In this case, the non-causal path containing $S$ must be considered for blocking. We again distinguish two visibility patterns.

*Visible case 1:* $S * \to X \to Y$. If $S$ is a collider on the non-causal path, then the path is inactive by default. If $S$ is a non-collider, conditioning on $S$ does not activate the path. Thus, $S$ can simultaneously witness visibility and serve as a valid adjustment variable.

One might ask whether $S$ could be a collider on one non-causal path from $X$ to $Y$ and a non-collider on another, in which case rules would fail. However, such a configuration cannot arise: during the graph learning process, this structure would be identified as non-amenable, and hence non-adjustable.

*Visible case 2:* This visibility pattern is even less likely in this context. In the corresponding true causal graph, the structure is non-amenable and therefore not adjustable. Consequently, $\mathcal{R}2$ applies. Together, $\mathcal{R}1$ and $\mathcal{R}2$ cover all amenable cases with a non-zero causal effect.

**Case 2: The causal effect from $X$ to $Y$ is zero.** If there is no path between $X$ and $Y$, or if all paths between them are inactive, then $X \perp\!\!\!\perp Y$ holds unconditionally.

If there exists an active non-causal path between $X$ and $Y$, conditioning on a set $\mathbf{Z} \subseteq \mathrm{MB}(X)$ with $\mathbf{Z} \cap \mathrm{PossDe}(X, \mathcal{P}) = \emptyset$ renders $X$ independent of all non-descendants. In this case, the first condition of $\mathcal{R}3$ applies.

For the case $X \leftrightarrow Y$, where no separating set exists, note that since $S \in \mathrm{MB}(X) \setminus (\{Y\} \cup \mathrm{Ch}(X, \mathcal{P}))$, any path or edge incident to $X$ from $S$ must have an arrowhead at $X$. By $\mathcal{R}3$, $X$ is therefore identified as a collider on the path from $S$ to $Y$, implying that $Y$ cannot be a descendant of $X$.

If no such $S$ exists, amenability itself becomes undecidable, corresponding to a non-adjustable case. Therefore, it is impossible to identify from observational data. That means that as long as we can identify from the observed data that the graph is amenable with respect to $(X, Y)$, then our rules can definitely identify it.

From the above analysis, if $\mathcal{R}1$, $\mathcal{R}2$ and $\mathcal{R}3$ are all inapplicable, then conditional independence and dependence relations among observed variables are insufficient to determine whether a causal effect from $X$ to $Y$ exists. $\qquad\square$

### B.6. Proof of Theorem 6

*Proof.* Assuming access to an oracle conditional independence test, the proposed structure learning algorithm can correctly recover the local causal structure around $X$. By Theorem 1, if a valid adjustment set exists, then we can find at least one valid adjustment set within $\mathrm{MB}(X)$. Furthermore, Theorems 2 and 3 characterize how such a valid adjustment set can be identified purely from features of the learned local structure. Together, Theorems 2 and 3 provide necessary and sufficient conditions for identifying a causal effect of $X$ on $Y$ that is inferable from the observed data, using only testable independence and dependence relationships among observed variables. Consequently, whenever a causal effect of $X$ on $Y$ is identifiable from observational data, the LCS algorithm is guaranteed to correctly identify and estimate this effect.

On the other hand, Theorems 4 and 5 establish that $\mathcal{R}3$ constitutes a necessary and sufficient condition for concluding the absence of a causal effect of $X$ on $Y$ that is inferable from the observed data. Accordingly, when no causal effect of $X$ on $Y$ exists that can be detected via observed (in)dependence relations, LCS correctly identifies this null effect.

Finally, if neither $\mathcal{R}1$, $\mathcal{R}2$ nor $\mathcal{R}3$ applies, then no conclusion of a causal effect of $X$ on $Y$ can be drawn solely from observational independence and dependence information. In this case, LCS appropriately returns an inconclusive result.

Above all, these results establish both the soundness and the completeness of the LCS algorithm for causal effect identification from observational data. $\square$

# C. Algorithm Description

## C.1. Examples of algorithms and rules based on the bnlearn network Mildew

We start from the Mildew network provided in the `bnlearn` benchmark and use it as a concrete graphical model to demonstrate our method. We randomly select two latent variables, `leldug_3` and `lemp_2`, and generate the corresponding PAG using functions from the `fastdag2pag` package. Based on this PAG, we present four examples to illustrate how the proposed algorithm operates and how the rules are applied during its execution.

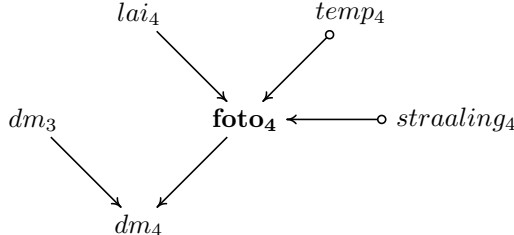

*Figure 8.* Induced subgraphs based on $MB^+(X)$ in Example 6.

**Example 6.** ***Input:*** *Target variable pair* $(X, Y) = (foto_4, udbytte)$ *and observed variable set* **O**.

1. ***Local structure learning around*** $X$.

$$MB(X) = \{lai_4, dm_4, dm_3, temp_4, straaling_4\},$$
$$Pa^*(X) = \{lai_4, temp_4, straaling_4\},$$
$$Pa(X) = \{lai_4\},$$
$$NCPa(X) = \{temp_4, straaling_4\},$$
$$Ch(X) = \{dm_4\}.$$

2. ***Rule application.*** *Using* $\mathcal{R}1$, *the variable* $S = temp_4$ *satisfies the conditional independence tests:*

$$temp_4 \not\perp\!\!\!\perp udbytte \mid \{lai_4\}, \quad foto_4 \perp\!\!\!\perp udbytte \mid \{lai_4, temp_4\}.$$

*Therefore, the adjustment set is*

$$\mathbf{Z} = \{lai_4\},$$

*and the procedure terminates with* $\Theta \leftarrow \theta$.

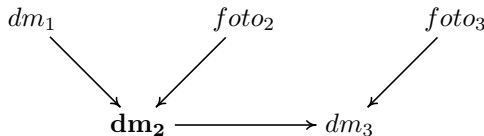

*Figure 9.* Induced subgraphs based on $MB^+(X)$ in Example 7.

**Example 7.** ***Input:*** *Target variable pair* $(X, Y) = (dm_2, dm_4)$ *and observed variable set* **O**.

1. *Local structure learning around $X$.*

$$MB(X) = \{dm_1, foto_2, dm_3, foto_3\},$$
$$Pa^*(X) = \text{Pa}(X) = \{dm_1, foto_2\},$$
$$Ch(X) = \{dm_3\}.$$

2. *Rule application. $\mathcal{R}1$ is not satisfied, so we proceed to $\mathcal{R}2$. $\mathcal{R}2$ holds since, in the local adjacency structure around $X$, the mark at the $X$-endpoint is always determined. Hence, the adjustment set is*

$$\mathbf{Z} = \{dm_1, foto_2\}.$$

*The causal effect is estimated and the procedure terminates with $\Theta \leftarrow \theta$.*

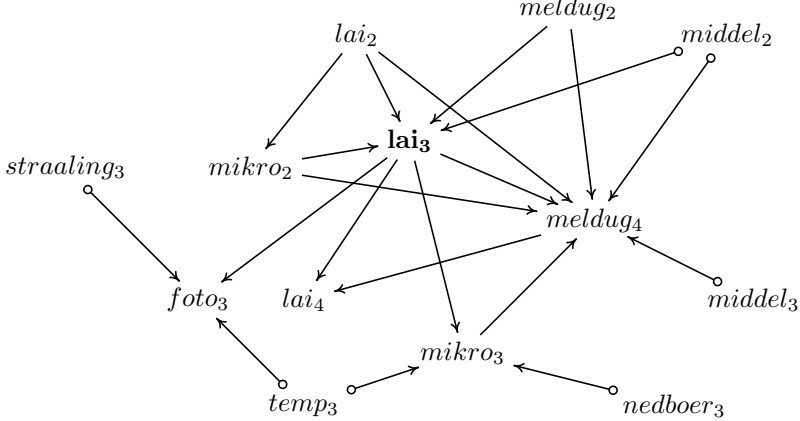

*Figure 10.* Induced subgraphs based on $MB^+(X)$ in Example 8.

**Example 8.** ***Input:*** *Target variable pair $(X, Y) = (lai_3, straaling_4)$ and observed variable set $\mathbf{O}$.*

1. *Local structure learning around $X$.*

$$MB(X) = \{mikro_2, lai_2, meldug_2, middel_2, foto_3, meldug_4, mikro_3, straaling_3, temp_3, nedboer_3, middel_3, lai_4\},$$
$$Pa^*(X) = \{mikro_2, lai_2, meldug_2, middel_2\},$$
$$Pa(X) = \{mikro_2, lai_2, meldug_2\},$$
$$NCPa(X) = \{middel_2\},$$
$$Ch(X) = \{foto_3, meldug_4, lai_4, mikro_3\}.$$

2. *Rule application. Using $\mathcal{R}3(i)$, we find that $lai_3$ and $straaling_4$ are independent. Hence, the algorithm terminates with*

$$\Theta \leftarrow 0.$$

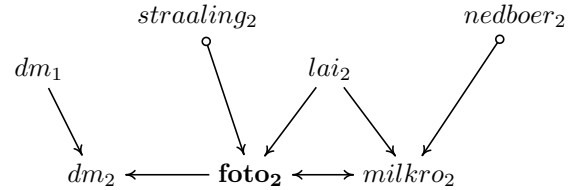

*Figure 11.* Induced subgraphs based on $MB^+(X)$ in Example 9.

**Example 9.** ***Input:*** *Target variable pair $(X, Y) = (foto_2, mikro_2)$ and observed variable set $\mathbf{O}$.*

1. ***Local structure learning around*** $X$**.**

$$MB(X) = \{lai_2, dm_2, dm_1, mikro_2, straaling_2, nedboer_2\},$$
$$Pa^*(X) = \{lai_2, straaling_2, mikro_2, nedboer_2\},$$
$$Pa(X) = \{lai_2\},$$
$$NCPa(X) = \{straaling_2\},$$
$$Ch(X) = \{dm_2\}.$$

2. ***Rule application.*** *Using* $\mathcal{R}3(i)$*, no conditioning set renders* $foto_2$ *and* $mikro_2$ *independent. Applying* $\mathcal{R}3(ii)$*, we find that*

$$straaling_2 \not\perp\!\!\!\perp foto_2 \quad but \quad straaling_2 \perp\!\!\!\perp mikro_2.$$

*The algorithm therefore terminates with*

$$\Theta \leftarrow 0.$$

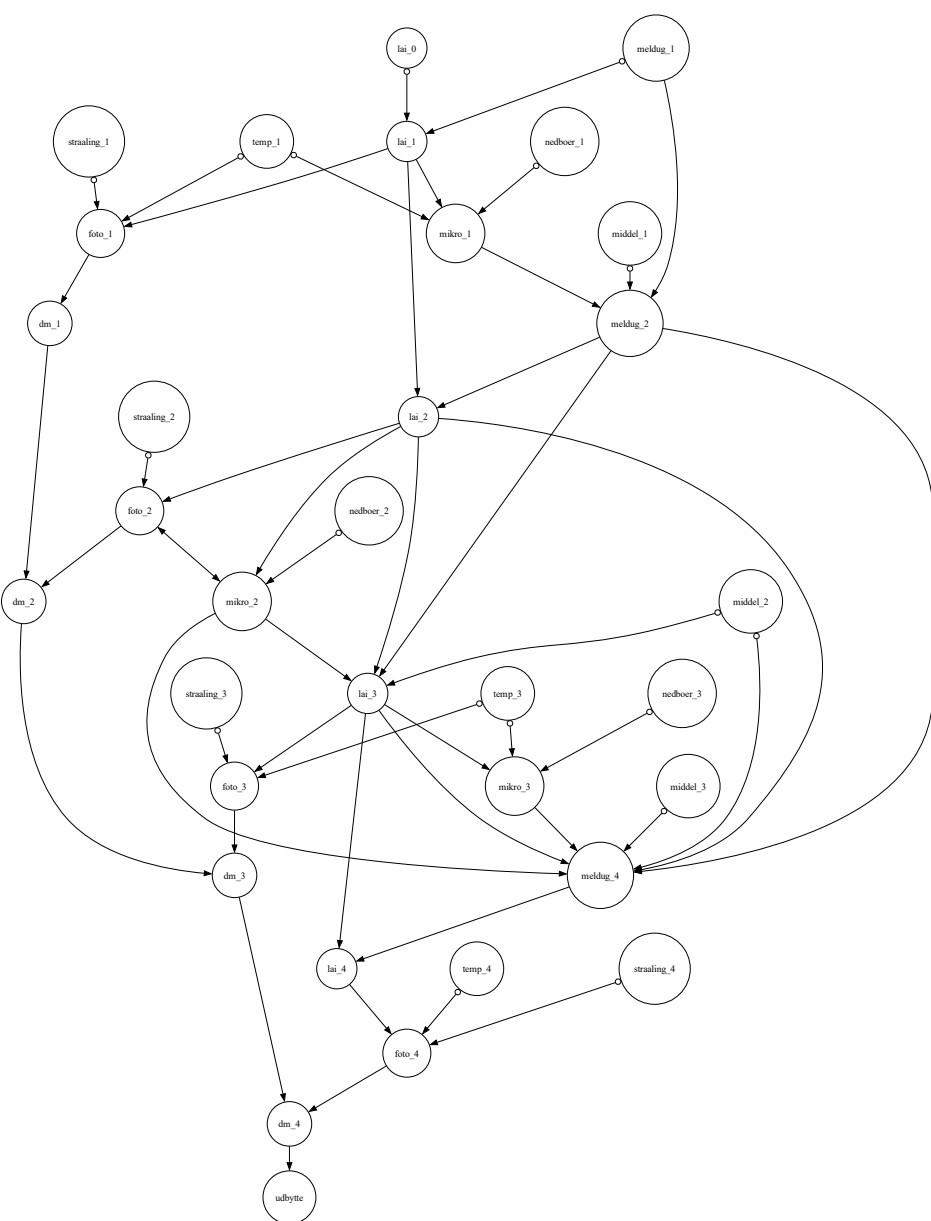

*Figure 12.* Example PAG (mildew network) with latent variables leldug_3, lemp_2.

## C.2. More details on the structure learning algorithm (Li et al., 2025b)

Here in Algorithm 2, we provide the details on the structure learning algorithm (Li et al., 2025b) that we used in our method, where Proposition 1, and Proposition 2 are in the following, while Lemma 1 is shown in Appendix B.1.

---

**Algorithm 2** Learning $\mathcal{P}_{MB^+(X)}$ Algorithm

---

**input** Target variable $X$, observed variables $\mathbf{O}$
**output** Learned partial ancestral graph $\mathcal{P}$ around $X$ (and thus $\mathcal{P}_{MB^+(X)}$)
  1: Initialize **Waitlist** $\leftarrow \{X\}$, **Donelist** $\leftarrow \emptyset$, $\mathcal{P} \leftarrow \emptyset$.
  2: **while** the stopping criteria in Proposition 1 are not met **do**
  3:     Let $V_i$ be the first variable in **Waitlist**.
  4:     Learn the Markov blanket $MB(V_i)$ from the data over $\mathbf{O}$.
  5:     Learn the local structure $\mathcal{L}_{V_i}$ over $MB^+(V_i)$.
  6:     Extract true edges and edge orientations from $\mathcal{L}_{V_i}$ using Lemma 1 and Proposition 2, and update $\mathcal{P}$.
  7:     Orient $\mathcal{P}$ using standard orientation rules.
  8:     Update **Waitlist** and **Donelist** accordingly.
  9: **end while**
 10: **return** $\mathcal{P}$

---

**Proposition 1** (**Stop Rules** (Li et al., 2025b)). *Let $X$ be the target variable of interest and* **Waitlist** *represent the collection of variables whose $\mathcal{L}$ will be learned. If any of the following rules are satisfied, the learned $\mathcal{P}_{MB^+(X)}$ is equivalent to the structure identified by global learning methods.*

$\mathcal{R}1$. *The edges among the variables of $MB^+(X)$ are all determined, i.e. , no circle present in the marks.*

$\mathcal{R}2$. *The* **Waitlist** *is empty.*

$\mathcal{R}3$. *All paths from each vertex in $MB^+(X)$, which include undirected edges (connected two vertices in $MB^+(X)$), are blocked by edges $*\rightarrow$.*

**Proposition 2.** *Let $\mathcal{L}_X$ be the inferred PAG over $MB^+(X)$. Let $V_i$ $(1 \leq i \leq |\mathrm{MB}(X)|)$ represent the vertices in $\mathrm{MB}(X)$. The following statements hold:*

$\mathcal{S}1$. *The unshielded collider triples (V-structures) $V_1 *\rightarrow X \leftarrow* V_2$ identified in $\mathcal{L}_X$ are consistent with those in the ground-truth PAG.*

$\mathcal{S}2$. *The uncovered collider paths $X *\rightarrow V_1 \leftrightarrow \cdots \leftarrow* V_i$ identified in $\mathcal{L}_X$ are consistent with those in the ground-truth PAG.*

