# OpenReview forum: "Local Covariate Selection for Average Causal Effect Estimation without Pretreatment and Causal Sufficiency Assumptions"
_ICML.cc/2026/Conference — ICML 2026 spotlight_

### Official Review · Reviewer_B559 · 2026-03-12

**Soundness:** 2
**Presentation:** 2
**Significance:** 2
**Originality:** 3
**Overall Recommendation:** 3
**Confidence:** 3

**Summary:**

This work studies the problem of covariate selection for estimating the average causal effect (ACE) of a treatment variable on an outcome from observational data. Many existing methods rely on strong assumptions such as causal sufficiency (all confounders are observed) or the pretreatment assumption (all candidate covariates are measured before treatment). However, these assumptions are often unrealistic in real-world applications where latent confounding and post-treatment variables may exist. To address these limitations, the paper proposes a Local Covariate Selection (LCS) method that identifies valid adjustment sets using only local causal structure information rather than learning the entire causal graph. The key idea is to exploit properties of the Markov boundary of the treatment variable to search for adjustment variables that allow unbiased estimation of the causal effect, even when latent variables or post-treatment variables are present.  Extensive experiments on synthetic datasets, benchmark Bayesian networks, and real-world data demonstrate that LCS can achieve more accurate causal effect estimation and lower computational cost compared with several existing local covariate selection methods.

**Compliance With Llm Reviewing Policy:**

Affirmed.

**Key Questions For Authors:**

**Questions:**
Once the pretreatment assumption is removed, how can we rule out the possibility that some selected variables are actually children of Y?

Could the authors provide the adjustment sets returned by each method on the Cattaneo2 dataset?

The current experiments seem somewhat self-contained. How can this approach be applied or validated in real-world scenarios?

**Limitations:**

yes

**Strengths And Weaknesses:**

**Strengths:**

1. The studied problem is well motivated.

2. LCS is sound and complete.

3. Experiments confirm the performance of LCS.

**Weaknesses:**

The theoretical claims are made under oracle conditional independence tests and standard assumptions such as Faithfulness. In practice, LCS depends on estimated local structure and CI tests that may be unstable in high dimensions. Since causal inference is a setting where the true causal effect is usually unobservable, it is difficult to diagnose where the estimation may go wrong.

The current experimental setup and validation seem somewhat self-evaluative. It would strengthen the work if the authors could demonstrate how this framework can be applied in real-world settings. The authors intentionally extend the feature space by adding a descendant of the treatment. However, in this setting it is unclear whether the reference interval derived from prior empirical studies can still be used as a meaningful benchmark. Introducing such a variable may alter the effective causal structure, potentially making the comparison with previous results less reliable.





The presentation of the paper needs improvement. For example, Lines 142–143 lack a space between two sentences.

---

> ### Author Rebuttal · Authors · 2026-03-31
>
> Thank you for your constructive feedback and for recognizing the motivation, soundness, completeness, and empirical effectiveness of LCS. We respond below.
>
> >**W1** The theoretical claims … under oracle CI tests and standard Faithfulness. … estimated local structure and CI tests unstable in high dimensions. … true causal effect unobservable … difficult to diagnose
>
> **A1:** We would like to clarify three points.
>
> **First**, oracle-CI and Faithfulness are used for theoretical guarantees; our experiments do not enforce them, and results show LCS remains effective when they are not perfectly held.
>
> **Second**, local learning approaches such as ours significantly reduce unnecessary CI tests. Compared to global approaches, this alleviates the instability caused by excessive CI testing in high-dimensional settings and mitigates minor Faithfulness violations [1].
>
> **Third**, we agree that the true causal effect is inherently unobservable in practice. However, this is a fundamental challenge in causal inference (Pearl, 2009; Spirtes et al., 2000) rather than a limitation specific to our method. Despite the difficulty of diagnosing estimation errors, principled data-driven methods remain widely used in practice [2–4]. Our work extends this line by relaxing pretreatment and causal sufficiency while maintaining soundness and completeness.
>
> [1] Isozaki T. A robust causal discovery algorithm against faithfulness violation. Information and Media Technologies, 2014.
>
> [2] Maathuis M H, et al. Predicting causal effects in large-scale systems from observational data. Nature methods, 2010.
>
> [3] Maathuis M H, et al. Estimating high-dimensional intervention effects from observational data. The Annals of Statistics, 2009.
>
> [4] Hernán M A, Robins J M. Causal inference: What If. 2010.
>
> >**W2&Q3** Experiment … self-evaluative. … applied or validated in real-world settings? … reference interval … still meaningful?
>
> **A2:** We would like to clarify the following points.
>
> **First**, our setup follows standard practice (Cheng et al., 2022, 2023; Li et al., 2025a), and the Cattaneo2 evaluation examines the practical behavior of LCS on real data.
>
> **Second**, adding a descendant of the treatment is intended to **explicitly violate the pretreatment assumption** and create a controlled real-data setting for testing robustness beyond pretreatment-based methods. This modification does not change the target estimand, namely the total causal effect of maternal smoking on birth weight. Instead, it allows us to directly examine whether a method can still estimate the total causal effect reliably when post-treatment variables are present among the observed covariates.
>
> **Third**, we use the reference interval from prior reported studies as a plausibility reference rather than the exact ground truth. In this modified setting, LCS returns the plausible estimate among the compared methods. For completeness, we also note that, without adding the descendant variable, the estimate returned by LCS remains within the same empirical reference range, indicating that its good performance is not specific to the modified setting.
>
> **Fourth**, this setup is also motivated by practice: post-treatment variables are often recorded together with baseline covariates, while it is difficult to verify from observational data alone whether a variable violates the pretreatment assumption. This is precisely why covariate selection without relying on pretreatment is practically important.
> We will revise the discussion of the real-world experiment to make this motivation clearer.
>
> >**Q1** … pretreatment removed … rule out … children of Y?
>
> **A3:** Without the pretreatment assumption, one might expect to exclude descendants of both $X$ and $Y$. However, by the second condition of the Generalized Adjustment Criterion (Definition 2), i.e., $Z \cap \mathrm{Forb}(X, Y) = \emptyset$, a valid adjustment set $Z$ cannot contain the descendants of any node on any possible directed path from $X$ to $Y$. Since $Y$ is a descendant of $X$ when a causal effect exists, descendants of $Y$ are also descendants of $X$. Thus, it suffices to exclude descendants of $X$. In practice, we achieve this via local structure learning (Li et al., 2025b), which is sufficient for ruling out variables that cannot belong to a valid adjustment set. We will clarify this in the revision.
>
> >**Q2** adjustment sets returned … on the Cattaneo2?
>
> **A4:** Yes. On the Cattaneo2 dataset, the adjustment set returned by LCS is \{*mmarried, nprenatal*\}, the set returned by LDP is \{*PHYPER, mhisp, fhisp, birthmonth*\}, the set returned by CEELS is \{*mrace, lbweight*\}, the set returned by LSAS is \{*lbweight*\}, and the set returned by EHS is \{*mmaried, mrace*\}. We will include these returned sets in the revised version to improve the transparency of the real-data comparison. Notably, LDP selects PHYPER, which is a descendant of $X$, leading to the most biased estimates, consistent with our analysis.

---

> > ### Author Rebuttal · Reviewer_B559 · 2026-04-03
> >
> > I appreciate the authors’ responses and clarifications. However, considering the remaining concerns regarding the practical usability of the proposed method, I will stand by my original rating.

---

> > > ### Author Response · Authors · 2026-04-04
> > >
> > > We thank you for the follow-up and for carefully revisiting our work. We would like to clarify that a central motivation of our method is precisely to **improve the practical usability of adjustment-based causal effect estimation**. In particular, LCS offers the following practical advantages:
> > >
> > > -- **Efficiency via local learning.** LCS is a local learning approach that significantly reduces the number of CI tests compared to global methods that search over all variables or require learning the full causal structure (Spirtes et al., 2000; Entner et al., 2013; Maathuis et al., 2015; Malinsky et al., 2017; Perković et al, 2018). By restricting the search to a local region, LCS greatly **reduces computational cost while still maintaining completeness guarantees**, making it more suitable for high-dimensional settings.
> > >
> > >
> > > -- **Relaxation of unrealistic assumptions.** LCS does not rely on either the causal sufficiency assumption or the pretreatment assumption, both of which are often unrealistic in practice (Malinsky et al., 2017; Maasch et al., 2024; Li et al., 2025). As summarized by Cheng et al. (2024), latent variables are common in real-world applications, while pretreatment assumptions are frequently adopted in existing data-driven causal effect estimation methods. These assumptions typically require strong prior causal knowledge: causal sufficiency assumes that no latent confounders exist among observed variables, while the pretreatment assumption requires knowledge of the causal ordering relative to the treatment and outcome. In contrast, **LCS does not require such prior knowledge, while still maintaining soundness and completeness**, making it applicable in more realistic scenarios than existing adjustment-set selection methods (Entner et al., 2013; Cheng et al., 2022; Maasch et al., 2024; Li et al., 2025).
> > >
> > >
> > > **To further support these practical advantages, we additionally evaluate our method on a real-world dataset.** Specifically, we conduct experiments on the widely used Jobs dataset (LaLonde, 1986; Imai & Ratkovic, 2014), which contains 2,936 samples with 10 variables and is commonly used to study the causal effect of a job training program on earnings for treated individuals. The reference ATT (Average Treatment Effect on the Treated) for this dataset is $886 (Imai & Ratkovic, 2014).
> > > As shown below, LCS achieves both higher accuracy and greater efficiency than the global method EHS, and higher accuracy than local methods such as LDP, CEELS, and LSAS, further demonstrating its effectiveness and practicality in real-world settings.
> > >
> > >
> > > |Method|ATT|Bias(%)|nTests|
> > > |-|-|-|-|
> > > |LCS|854.528|3.55%|303|
> > > |EHS|930.606|5.04%|1732|
> > > |LSAS|989.346|11.67%|176|
> > > |CEELS|683.406|22.87%|183|
> > > |LDP|-13597.588| 1635.98%|61|
> > >
> > >
> > > We hope these clarifications further demonstrate the practical usability of LCS. We thank the reviewer again for the thoughtful feedback and would be happy to discuss any remaining questions.
> > >
> > >
> > >
> > >
> > >
> > > - LaLonde R J. Evaluating the econometric evaluations of training programs with experimental data. Am. Econ. Rev., 1986.
> > > - Imai K, Ratkovic M. Covariate balancing propensity score. JRSSB, 2014.
> > > - Spirtes P, Glymour C N, Scheines R. Causation, prediction, and search. MIT press, 2000.
> > > - Maathuis M H, Kalisch M, Bühlmann P. Estimating high-dimensional intervention effects from observational data. The Annals of Statistics, 2009.
> > > - Entner D, Hoyer P, and Spirtes P. Data-driven covariate selection for nonparametric estimation of causal effects. AISTAT, 2013.
> > > - Maathuis M H, & Colombo D. A generalized back-door criterion. The Annals of Statistics, 2015.
> > > - Malinsky D and Spirtes P. Estimating bounds on causal effects in high-dimensional and possibly confounded systems. IJAR, 2017.
> > > - Perković E, et al. Complete graphical characterization and construction of adjustment sets in Markov equivalence classes of ancestral graphs. JMLR, 2018.
> > > - Cheng, D., Li, J., Liu, L., Zhang, J., Liu, J., and Le, T. D. Local search for efficient causal effect estimation. IEEE Transactions on Knowledge and Data Engineering, 2022.
> > > - Maasch J, et al. Local discovery by partitioning: Polynomial-time causal discovery around exposure-outcome pairs. UAI, 2024.
> > > - Cheng, D., Li, J., Liu, L., Liu, J., & Le, T. D. Data-driven causal effect estimation based on graphical causal modelling: A survey. ACM Computing Surveys, 2024.
> > > - Li, Z., Guo, X., Xie, F., Zeng, Y., Zhang, H., and Geng, Z. Local learning for covariate selection in nonparametric causal effect estimation with latent variables. In Advances in Neural Information Processing Systems, 2025.

---

### Official Review · Reviewer_uzpJ · 2026-03-13

**Soundness:** 3
**Presentation:** 3
**Significance:** 3
**Originality:** 3
**Overall Recommendation:** 3
**Confidence:** 2

**Summary:**

The paper develops a method for selecting conditioning covariates to enable unbiased estimation of treatment effect.
In contrast to the literature, the proposal relies only on local structure rather than the global structure, and does not require pretreatment variables or causal sufficiency.
The method is shown to be sound and complete, and it has lower computational complexity when the extended Markov blanket is small.

**Compliance With Llm Reviewing Policy:**

Affirmed.

**Final Justification:**

I thank the authors’ rebuttal. I apologize that this is not my area of expertise.

**Key Questions For Authors:**

See strengths and concerns.

**Limitations:**

See strengths and concerns.

**Strengths And Weaknesses:**

Strengths.

The problem is important, since unbiased estimation is a central goal in causal inference, and identifying a valid adjustment set is critical for that purpose.

Existing methods seem to face challenges from both computational complexity and strong theoretical assumptions.
This paper seems to make contributions in both directions.

1.  On the methodological side, Theorems 2 and 4 characterize sufficient conditions for a valid adjustment set. Theorem 5 establishes the soundness and completeness.

2. On the computational side, although the complexity still scales exponentially with the size of the extended Markov blanket, this is an improvement over methods whose complexity scales exponentially with the total number of nodes.

Concerns.

I am not very experienced in this area, and therefore do not have any major concerns about the paper.

---

> ### Author Rebuttal · Authors · 2026-03-30
>
> Thank you for your positive and encouraging assessment. We especially appreciate your recognition of the importance of the problem, the theoretical value of the soundness/completeness results, and the computational benefit of restricting the search to the extended Markov blanket rather than the full variable set.
>
> We would like to further emphasize that a key challenge of this problem is to perform local covariate selection without relying on either the pretreatment assumption or causal sufficiency, while still retaining soundness and completeness. What we believe is particularly novel in our work is that it addresses all three aspects simultaneously: it avoids these two strong assumptions, remains fully local, and provides sound and complete guarantees within the adjustment-based identification setting.
>
> In the revision, we will further improve the presentation and clarify the main points raised during review. We hope these clarifications and planned revisions will further strengthen your overall assessment of the paper.

---

> > ### Author Rebuttal · Reviewer_uzpJ · 2026-04-03
> >
> > I am not very familiar with this area and was unable to raise critical points in the initial review. I thank the authors for still providing a response :)

---

> > > ### Author Response · Authors · 2026-04-03
> > >
> > > Thank you. We truly appreciate your acknowledgement that the concerns have been fully resolved.
> > >
> > > We respectfully note that this updated assessment appears more positive than the current numerical score. If appropriate, and if this better reflects your current view of the paper, we would be grateful if you could consider revising the score accordingly.

---

### Official Review · Reviewer_4kQZ · 2026-03-13

**Soundness:** 2
**Presentation:** 2
**Significance:** 2
**Originality:** 2
**Overall Recommendation:** 3
**Confidence:** 4

**Summary:**

This paper presents a local method for selecting adjustment sets to estimate causal effects (LCS). Unlike existing methods, it does not assume pretreatment variables or no latent confounders. The key idea is to use the treatment’s Markov blanket to find valid adjustment sets. The authors present three identification rules and prove the method is sound and complete. Experiments show LCS achieves lower error with fewer independence tests than existing methods.

**Compliance With Llm Reviewing Policy:**

Affirmed.

**Final Justification:**

Thanks to the authors for trying to address my main concern related to the observed adjustment sets. The main argument is that this limitation is not specific to the proposed method, and the authors cite PC-style methods and IV-based methods. However, I am still not very clear about the positioning of the paper, as PC-style methods assume causal sufficiency, so observed adjustment sets make sense, while IV-based methods account for hidden confounders. Therefore, I will keep my initial assessment score.

**Key Questions For Authors:**

The method only uses observed variables for adjustment. Does it mean that the method cannot deal with datasets with hidden variables at all or do I miss the point?

Can the authors clarify the limits of their identifiability claims in the presence of latent variables and the scope of the method’s applicability?

The real-world experiment adds a descendant variable artificially. Can the authors show a case where such violations occur naturally in real data?

**Limitations:**

yes

**Strengths And Weaknesses:**

Strengths:

Well structured paper with clear motivation and helpful figures.

LCS achieves lower estimation error and requires fewer independence tests in experiments.

Weaknesses:

Limited to adjustment sets within observed variables, so it cannot handle cases requiring methods like IVs

Finite-sample effects are not discussed.

Real-world evaluation is limited to one dataset with artificially introduced collider bias.

---

> ### Author Rebuttal · Authors · 2026-03-31
>
> Thank you for your comments and for recognizing the motivation, structure, and empirical efficiency of our paper. We hope this addresses your concerns.
> >**W1&Q1** Limited to adjustment sets within observed variables … cannot handle cases...like IVs or datasets with hidden variables...?
>
> **A1:** LCS is designed for **adjustment-based causal effect identification** and does **not** assume the absence of hidden variables. In our setting, hidden variables are represented by bidirected edges in the MAGs. Our goal is to determine whether the causal effect is identifiable via an observed adjustment set satisfying the GAC. Therefore, searching over observed adjustment variables does not mean that LCS cannot handle hidden variables. This is different from other identification strategies, such as IV-based methods, which address a different identification setting.
>
> >**W2** Finite-sample effects...
>
> **A2:**  Section 5.1 already evaluates finite-sample behavior (1K–10K samples for random graphs; 5K–15K for bnlearn). Following your suggestion,  we additionally ran random graph experiments with 500–2K samples, summarized below.
>
> Results show a consistent trend: LCS maintains the best accuracy–efficiency trade-off, achieving lower relative error while requiring far fewer CI tests than global methods (EHS) and fewer than most local baselines. This confirms that LCS remains robust even in smaller-sample regimes. We will include these results in the revision.
>
> |Nodes|Size|LCS(RE,nT)|LSAS|CEELS|LDP|EHS|
> |-|-|-|-|-|-|-|
> |20|500|0.257,51|0.295,452|0.332,297|0.384,117|0.537,62247|
> ||1K|0.194,119|0.243,886|0.315,418|0.388,128|0.477,71338|
> ||2K|0.179,156|0.213,1189|0.279,548|0.295,139|0.446,78542|
> |30|500|0.227,100|0.286,815|0.345,490|0.333,187|—|
> ||1K|0.196,342|0.221,1881|0.250,669|0.316,202|—|
> ||2K|0.150,883|0.191,4214|0.185,911|0.159,208|—|
> |40|500|0.274,182|0.298,851|0.319,587|0.410,221|—|
> ||1K|0.172,535|0.282,3957|0.260,797|0.344,246|—|
> ||2K|0.133,1357|0.285,8468|0.201,1118|0.304,266|—|
> |50|500|0.387,218|0.370,1875|0.380,793|0.271,258|—|
> ||1K|0.197,1114|0.315,6018|0.353,1059|0.387,295|—|
> ||2K|0.122,2750|0.278,8927|0.242,1577|0.209,325|—|
>
> Note: EHS is omitted in some tables due to prohibitive exhaustive-search cost.
>
>
> >**W3&Q3** Real-world evaluation...artificially introduced collider bias. A case where such violations occur naturally in real data?
>
> **A3:** We would like to clarify the following points.
>
> (1). Adding a descendant explicitly violates the pretreatment assumption in a controlled way, allowing us to test robustness when post-treatment variables are present. In this modified setting, LCS yields the plausible estimate among the compared methods with moderate testing cost. For completeness, without the added descendant, the estimate returned by LCS also remains within the reference ATE range, indicating that its good performance is not specific to the modified setting.
>
> (2). In practice, it is difficult to determine from observational data whether a variable violates the pretreatment assumption, since this typically requires unavailable causal knowledge. This is one motivation for studying covariate selection without relying on pretreatment. Meanwhile, post-treatment variables are often recorded with baseline covariates. For example, in a clinical study of the effect of a medication on body weight, variables such as appetite or caloric intake may be measured after treatment and included in the dataset, making pretreatment violations possible.
>
> (3). To further validate practical effectiveness, we evaluated LCS on the Jobs dataset (LaLonde, 1986; Imai & Ratkovic, 2014), which contains 2,936 samples with 10 variables and is used to study the causal effect of a job training program on earnings for treated individuals. The reference ATT (Average Treatment effect on the Treated) for this dataset is $886 (Imai & Ratkovic, 2014). As shown below, LCS achieves both higher accuracy and greater efficiency than the global method EHS, and higher accuracy than the local methods LDP, CEELS, and LSAS.
>
> |Method|ATT|Bias(%)|nTests|
> |-|-|-|-|
> |LCS|854.528|3.55%|303|
> |EHS|930.606|5.04%|1732|
> |LSAS|989.346|11.67%|176|
> |CEELS|683.406|22.87%|183|
> |LDP|-13597.588| 1635.98%|61|
>
> - LaLonde R J. Evaluating the econometric evaluations of training programs with experimental data. AER, 1986.
> - Imai K, Ratkovic M. Covariate balancing propensity score. JRSSB, 2014.
>
> >**Q2** ...clarify the limits and the scope...?
>
> **A4:** Regarding scope, LCS applies to adjustment-based identification under observable CI/dependence information, even with hidden variables and without the pretreatment assumption. Its main limitation is that the guarantee applies only to effects identifiable via an observed valid adjustment set. If LCS cannot determine the effect, it is non-identifiable in this adjustment-based sense. Such cases may require other strategies, such as IV-based identification, and LCS correctly returns them as non-identifiable.

---

> > ### Author Rebuttal · Reviewer_4kQZ · 2026-04-02
> >
> > Thank you for the answer to my first question regarding observed adjustment sets. I understand the discussion and the difference between LCS and IV-based methods. However, my main concern is that the range of applications of LCS may be limited, as it requires the existence of a subset of observed variables that can serve as an adjustment set. While hidden variables are acknowledged in LCS (via MAG), the implicit assumption is that they cannot be part of the adjustment set. I will keep my current score for now.

---

> > > ### Author Response · Authors · 2026-04-03
> > >
> > > Thank you for the follow-up and for taking another look at our work. We agree that LCS is limited to causal effects that are identifiable via an **observed adjustment set**, and we will make this scope more explicit in the revision.
> > >
> > > However, we would like to clarify an important distinction. The concern, as we understand it, **is better framed as a limitation of adjustment-based identification, which requires the existence of a valid observed adjustment set. It is therefore not a limitation specific to LCS in handling hidden variables,** since such hidden variables are explicitly allowed in our MAG setting. Since hidden variables are unobserved, they cannot be used directly for estimation; rather, their role is to determine whether identification through some sets of observed variables is possible (Pearl, 1995; Maathuis & Colombo, 2015; Perković et al., 2018).
> > >
> > > More broadly, covariate adjustment remains one of the most widely used and practically relevant strategies in observational causal inference (Pearl, 2009; Spirtes et al., 2000). In this sense, requiring an **observed** adjustment set is not a special restriction introduced by LCS, but a basic feature of adjustment-based identification itself. Likewise, **IV-based methods also require the existence of observed variables that satisfy the instrumental-variable assumptions**, that is, valid observed IVs, although under a different identification framework (Hernán & Robins, 2006; Imbens & Rubin, 2015; Silva & Shimizu, 2017). Therefore, the main scope limitation of LCS is not the presence of hidden variables per se. Instead, LCS is specialized to **adjustment-based identification** and does not aim to cover effects identifiable only through non-adjustment strategies.
> > >
> > >
> > > Moreover, in adjustment-based methods, one of the central questions is precisely whether a valid observed covariate adjustment set exists so that estimation can proceed directly from observational data (Entner et al., 2013; Cheng et al., 2022; Li et al., 2025). Within this setting, our contribution is to identify valid adjustment sets from observational data **without relying on either the pretreatment assumption or causal sufficiency**, while remaining **fully local** and providing **sound and complete guarantees**. Compared with existing global adjustment-set selection methods, LCS avoids full causal structure learning and exhaustive global search; compared with existing local methods, it achieves completeness without relying on the pretreatment assumption.
> > >
> > > We thank you for pointing this out and will revise the paper to clarify the scope and limitations of LCS more explicitly.
> > >
> > >
> > >
> > > - Pearl J. Causal Diagrams for Empirical Research. Biometrika, 1995.
> > > - Maathuis M H, & Colombo D. A generalized back-door criterion. The Annals of Statistics, 2015.
> > > - Perković E, Textor, J., Kalisch, M., & Maathuis, M. H. Complete graphical characterization and construction of adjustment sets in Markov equivalence classes of ancestral graphs. JMLR, 2018.
> > > - Pearl, J. Causality. Cambridge university press, 2009.
> > > - Spirtes P, Glymour C N, Scheines R. Causation, prediction, and search. MIT press, 2000.
> > > - Hernán, M. A. and Robins, J. M. Instruments for causal inference: an epidemiologist’s dream? Epidemiology, pp.360–372, 2006.
> > > - Imbens, G. W. and Rubin, D. B. Causal inference for statistics, social, and biomedical sciences: An introduction. Cambridge University Press, 2015.
> > > - Silva R, Shimizu S. Learning instrumental variables with structural and non-gaussianity assumptions. JMLR, 2017.
> > > - Entner D, Hoyer P, and Spirtes P. Data-driven covariate selection for nonparametric estimation of causal effects. AISTAT, 2013.
> > > - Cheng, D., Li, J., Liu, L., Zhang, J., Liu, J., and Le, T. D. Local search for efficient causal effect estimation. IEEE Transactions on Knowledge and Data Engineering, 2022.
> > > - Li, Z., Guo, X., Xie, F., Zeng, Y., Zhang, H., and Geng, Z. Local learning for covariate selection in nonparametric causal effect estimation with latent variables. In Advances in Neural Information Processing Systems, 2025.

---

### Official Review · Reviewer_wrf1 · 2026-03-16

**Soundness:** 3
**Presentation:** 3
**Significance:** 3
**Originality:** 3
**Overall Recommendation:** 5
**Confidence:** 2

**Summary:**

The paper studies the problem of covariate selection for causal estimation.
The authors adopt a local search approach, which only requires learning some confounders locally and then provides identification and estimation of the causal effect.
They identify a common limitation of previous approaches: the pretreatment assumption. Moreover, works that try to waive this assumption provide techniques that are not sound (always identify correct causal effects) and complete (they identify causal effects whenever they are identifiable).

The authors theoretically characterize a local boundary set (neighborhood of the treatment in the causal graph) and its properties such that the causal effect is identifiable.
Subsequently, they develop an algorithm based on these properties to identify sets of variables that allow us to estimate causal effects. They prove that their algorithm is both sound and complete, in contrast to all prior works.
Finally, they provide experimental results on both synthetic and real-world datasets that indicate how their approach improves over previous methods.

**Compliance With Llm Reviewing Policy:**

Affirmed.

**Final Justification:**

I believe the paper addresses an important gap in the literature. The paper characterizes the identifiability of causal effect assuming knowlegde of the conditional independencies and dependencies among the observed variables, which is an assumption widely used in the literature to the best of my knowledge. It also provides an algorithm to identify (and estimate) the causal effect, whenever this is possible.

My prior concerns were mainly based on my understanding of the result, since the explanation-discussion at some points of the paper seemed inconsistent to the theorem statements. During the rebuttal, the authors clarified these points and convinced me for the soundness of their result. Given this, I beleive the paper closes an important gap in the literature and suggest acceptance.

**Key Questions For Authors:**

Major:
1. Am I right to understand that the formal definition for a valid adjustment set is definition 2? Could you explicitly mention this in the paper if it is, or provide a formal definition for this concept if it isn't?

2. In Theorem 1, you mention that " a subset of O is a valid adjustment set for estimating the total causal effect of X on Y if and only if there exists a subset of MB(X) that is a valid adjustment set for X and Y ." Do you mean to say that, *there exists a subset of O* that is a valid adjustment set ...? I think your interpretation of the theorem right below means to say that, but currently the theorem is a bit confusing; the opposite direction it could read like "if there exists a subset of MB(X) that is a valid adjustment set for X and Y, then *any* subset of O is a valid adjustment set for estimating the total causal effect of X on Y". Is that what you mean to say?

3. In Theorem 6, you write that "If none of these rules applies, however, then based on the testable conditional independencies and dependencies among the observed variables, *the LCS algorithm* cannot determine whether X has a causal effect on Y." However, later you interpret this theorem as saying that *no* algorithm with access only to these dependencies can determine the causal effect? Could you explain this inconsistency?

Minor:

4. I repeat my question for the runtime of LCS algorithm here. Could you discuss the exponential dependence on the size of the Markov Blanket of $X$?

5. In the experiments section, on the performance of LCS on Random Graphs, the RE vs Sample size experiment for *child* does not show the LDP (green line). Is it because it has very bad performance, so you had to truncate it? Or does it coincide with some other line?

**Limitations:**

yes

**Strengths And Weaknesses:**

Strengths:
1. The paper is well-written; it states the problem clearly and discusses prior literature extensively.
2. The paper fills an important research gap in causal effect estimation by providing a sound and complete method for identifying causal effects based on the local structure of the causal graphs.

Weaknesses:
1. The runtime of the proposed algorithm is exponential in the size of the boundary of the treatment variable $X$. The authors don't discuss this phenomenon; is it necessary, or could the algorithms improve?

I have some clarification questions that are important for my understanding of the results, which I mention in the next section. I am willing to raise my score if these questions are addressed.

---

> ### Author Rebuttal · Authors · 2026-03-30
>
> Thank you for your thoughtful comments and for recognizing that the paper is well written and addresses an important gap in local causal effect estimation. We appreciate your careful reading and are happy to clarify the points below.
>
> >**Q1** Is the formal definition of a valid adjustment set Definition 2?
>
> **A1:** Yes. In our paper, Definition 2 is intended to be the formal definition of a valid adjustment set. More precisely, a set $Z$ is a valid adjustment set for unbiased estimation of the total causal effect of $X$ on $Y$ if $Z$ satisfies Definition 2 relative to $(X, Y)$.  We have clarified this explicitly in the revised version.
>
> > **Q2** In Theorem 1, you mention that " a subset of O is a valid adjustment set …" Do you mean that there exists a subset of O that is a VAS? The theorem is a bit confusing …
>
> **A2:** Thank you for pointing this out. The intended meaning of Theorem 1 is the following existence equivalence: $\exists Z_1\subseteq \mathbf{O}$ such that $Z_1$ is a valid adjustment set relative to $(X, Y)$ **if and only if** $\exists Z_2\subseteq \mathbf{MB}(X)$ such that $Z_2$ is a valid adjustment set relative to $(X, Y)$. In other words, a valid adjustment set exists in the full observed set $\mathbf{O}$ if and only if one exists within $\mathbf{MB}(X)$. We have modified Theorem 1 accordingly.
>
>
> >**Q3** In Theorem 6, you write that "If none of these rules applies … the LCS algorithm cannot determine whether X has a causal effect on Y." However, later you interpret this theorem as saying that no algorithm with access only to these dependencies can determine the causal effect? Could you explain this inconsistency?
>
>
> **A3:** Thank you for raising this important point. We would like to clarify the following. Theorem 5 characterizes the identifiability boundary: if none of R1–R3 applies, then the causal effect is not identifiable from the testable conditional independencies and dependencies among the observed variables. Theorem 6 states that LCS is sound and complete with respect to this boundary. Therefore, when LCS cannot determine whether $X$ has a causal effect on $Y$, the point is not merely that the algorithm fails, but that the effect cannot be determined from this class of observable conditional independence/dependence information alone. We have revised the interpretation of Theorem 6 accordingly.
>
>
> >**Q4&W1** discuss the exponential dependence on the size of the Markov Blanket of X?
>
>
> **A4:** Thank you for this helpful suggestion. The worst-case exponential dependence comes from the completeness requirement: to ensure that no valid adjustment set within the local boundary is overlooked, one must in general consider candidate subsets within that boundary. This is also the case for other complete methods. For example, the global method EHS has exponential complexity in the number of all observed variables, while the local method LSAS has exponential complexity in the size of $\mathbf{MB}(Y)$; both rely on the pretreatment assumption. By contrast, the key advantage of LCS is that it restricts this exponential search to the local boundary $\mathbf{MB}(X)$, rather than the full observed variable set, while also avoiding the pretreatment assumption. We have clarified this point in the revised version.
>
>
>
> >**Q5** In the experiments section, on the performance of LCS on Random Graphs, the RE vs Sample size experiment for child does not show the LDP (green line). Is it because it has very bad performance, so you had to truncate it? Or does it coincide with some other line?
>
> **A5:** The LDP curve in that panel does not coincide with another method. It is not shown because its RE is substantially larger than those of the other methods, so the corresponding values fall far outside the displayed plotting range. We will clarify this in the revised version.

---

> > ### Author Rebuttal · Reviewer_wrf1 · 2026-04-02
> >
> > Thank you for your kind clarifications. Given these explanations, I understand the results and believe they consitute an important addition to the current literature. I will adjust my scores accordingly.

---

> > > ### Author Response · Authors · 2026-04-03
> > >
> > > Thank you very much for your kind feedback and for acknowledging our clarifications. We are pleased that your concerns have been fully addressed and that the explanations helped improve the understanding of our results. We truly appreciate your recognition of the contribution of our work, as well as your willingness to adjust your score accordingly.

---

### Decision · Program_Chairs · 2026-04-30

**Decision:**

Accept (spotlight)

**Comment:**

This paper considers the case of finding valid adjustment sets for treatment effect estimation and covariates are not necessarily pre-treatment and there could be unobserved confounding. One approach is find a the entire PAG (Partial Ancestral Graph) which is an equivalence class of true MAGs (Maximal Ancestral graphs) - representation of ancestral causal DAGs with confounding effects (that cannot be removed under observations).

Authors show such search for such adjustment sets can be confined to Markov Blanket of the treatment variable in the underlying MAG. Further authors show multiple conditions under which adjustments can be identified when some conditional independence assumptions involving a variable from Markov Blanket of X , the candidate valid adjustment, Treatment X and outcome Y are met.

Authors propose to use LCS algorithm (local covariate selection algorithm) to actually do these CI tests that these conditions locally around treatment X. Authors also show a completeness results as well. The paper also makes effort to distinguish their results from other prior work that use PAGs. Authors compare it on Cattaneo dataset and also included the Jobs dataset (during rebuttal) to demo their results.

Handling reviewer concerns (since some of them were negative and lukewarm)
 1) One reviewer admitted to not being familiar in this area but gave a low score - I am disregarding this score.
 2) Another reviewer mistakes the setting to assume causal sufficiency due to PC style algorithms. This is a misunderstanding as the authors have clearly clarified as well.

Given the strength of the results and some experimental validation that show the algorithm requiring lesser CI tests overall - this is a solid contribution.

Recommendation: Accept.